# Existence of Quantum Pharmacology in Sartans: Evidence in Isolated Rabbit Iliac Arteries

**DOI:** 10.3390/ijms242417559

**Published:** 2023-12-16

**Authors:** Laura Kate Gadanec, Jordan Swiderski, Vasso Apostolopoulos, Kostantinos Kelaidonis, Veroniki P. Vidali, Aleksander Canko, Graham J. Moore, John M. Matsoukas, Anthony Zulli

**Affiliations:** 1Institute for Health and Sport, Victoria University, Melbourne, VIC 3030, Australia; laura.gadanec@live.vu.edu.au (L.K.G.); jordan.swiderski@live.vu.edu.au (J.S.); vasso.apostolopoulos@vu.edu.au (V.A.); 2Immunology Program, Australian Institute for Musculoskeletal Science (AIMSS), Melbourne, VIC 3021, Australia; 3NewDrug PC, Patras Science Park, 26 504 Patras, Greece; k.kelaidonis@gmail.com; 4Institute of Nanoscience and Nanotechnology, National Centre for Scientific Research “Demokritos”, Ag. Paraskevi, 153 41 Athens, Greece; v.vidali@inn.demokritos.gr (V.P.V.); alexcanko1993@gmail.com (A.C.); 5Pepmetics Inc., 772 Murphy Place, Victoria, BC V6Y 3H4, Canada; mooregj@shaw.ca; 6Department of Physiology and Pharmacology, Cumming School of Medicine, University of Calgary, Calgary, AB T2N 4N1, Canada; 7Department of Chemistry, University of Patras, 265 04 Patras, Greece

**Keywords:** angiotensin II, angiotensin II receptor blockers, bisartans, inverse agonism, sartans

## Abstract

Quantum pharmacology introduces theoretical models to describe the possibility of ultra-high dilutions to produce biological effects, which may help to explain the placebo effect observed in hypertensive clinical trials. To determine this within physiology and to evaluate novel ARBs, we tested the ability of known angiotensin II receptor blockers (ARBs) (candesartan and telmisartan) used to treat hypertension and other cardiovascular diseases, as well as novel ARBs (benzimidazole-*N*-biphenyl tetrazole (ACC519T), benzimidazole-*bis*-*N*,*N*′-biphenyl tetrazole (ACC519T(2)) and 4-butyl-N,N0-bis[[20-2Htetrazol-5-yl)biphenyl-4-yl]methyl)imidazolium bromide (BV6(K^+^)_2_), and nirmatrelvir (the active ingredient in Paxlovid) to modulate vascular contraction in iliac rings from healthy male New Zealand White rabbits in responses to various vasopressors (angiotensin A, angiotensin II and phenylephrine). Additionally, the hemodynamic effect of ACC519T and telmisartan on mean arterial pressure in conscious rabbits was determined, while the ex vivo ability of BV6(K^+^)_2_ to activate angiotensin-converting enzyme-2 (ACE2) was also investigated. We show that commercially available and novel ARBs can modulate contraction responses at ultra-high dilutions to different vasopressors. ACC519T produced a dose-dependent reduction in rabbit mean arterial pressure while BV6(K^+^)_2_ significantly increased ACE2 metabolism. The ability of ARBs to inhibit contraction responses even at ultra-low concentrations provides evidence of the existence of quantum pharmacology. Furthermore, the ability of ACC519T and BV6(K^+^)_2_ to modulate blood pressure and ACE2 activity, respectively, indicates their therapeutic potential against hypertension.

## 1. Introduction

Placebo groups are an integral component in randomized controlled drug trials to allow differentiation between the specific effects of a pharmaceutical from non-specific changes in symptomology caused by a pharmacologically inactive substance [1]. Therefore, the effects observed in the experimental drug group are required to be substantial in comparison to those in the placebo group to establish drug efficacy and justify its use [1]. However, reports from drug trials demonstrate that both statistically and clinically significant improvements in symptoms have been observed in placebo groups, a concept defined as the “placebo effect” [1]. Incredibly, in contrast to the placebo effect, adverse or even toxic consequences have also been reported in placebo groups, a phenomenon known as the “nocebo effect” [2]. The paradox of these phenomena lies within the widely accepted dogma that an inert substance by definition is unable to produce an effect; neither positive nor negative [2]. However, there is compelling evidence from pharmacological studies indicating that placebos mimic the action of drugs through shared biochemical pathways by associating and activating the same receptor [3]. In an attempt to explain the complex underlying mechanisms of the placebo effect, multiple theories, such as natural course of disease, fluctuations in severity of symptoms, response bias involving patient reporting, other concurrent treatments, psychosocial expectation responses, and quantum physics and mechanics, have been hypothesized [2,4]. The latter suggests that information entanglement driven by human intention can alter the physical and biochemical properties and subsequently the physiological response of a placebo on a molecular and atomic level [4].

The notion that quantum mechanics can play a non-trivial role in biology has captured the attention of researchers for over a century. Initially, it was thought that the classical nature of biology and the realm of quantum mechanics were too distinct to interact, due to the presence of a complex “warm and noisy” environmental influences [5]. However, various biological phenomena, including energy transfer and light harvesting during photosynthesis [6], avian magnetoreception [7], electron tunnelling during DNA mutation [8], and electron transfer in the vibration theory of olfactory receptor activation [9], are linked to the effects of quantum mechanics [10].

Since the late 1970s, a large body of experimental work has been dedicated to understanding the bioactivity of compounds diluted beyond Avogadro’s constant, in which there would be no probability of the original compound(s) being present. Termed ultra-high dilutions, various studies have observed counterintuitively that these diluted perpetrations are able to induce physiological action and demonstrate molecular-level effects [11,12,13]. Akin to placebo, an interesting and challenging aspect of this phenomenon in research is that an inert diluted substance (i.e., water), which no longer contains pharmacological active compounds can activate biochemical pathways [14]. Although the mechanism of this phenomenon has never been elucidated, researchers have theorized the involvement of electromagnetic wave emissions to potentially induce bioactive effects [13]. Insufficient validation and the speculative nature of this theory have drawn heavy criticism primarily due to the implausibility of any substance remaining in the solution to induce an effect [15]. However, such an argument relies heavily on the assumption of the classical nature of biology and chemistry. The application of quantum mechanics and theoretical chemistry to describe biological phenomena represents a growing interest of research and presents a foundation for explaining the potential mechanisms of ultra-high dilutions [16].

Quantum mechanics described matter (i.e., electrons [17], atoms [18], and molecules [19]) as containing both a particle and a wave, which allows them to exhibit an interference pattern when passed through double slits [20]. Even when they pass individually, one by one, over time, an interference pattern emerges, indicating that particles are entangled with one another to contribute to the interference pattern [20]. This behavior of particles is attributed to quantum coherences and allows particles to coherently interact with each other to form a single state that may contain a physiological role in various biological processes [10,21]. From this understanding, a new model for liquid water has emerged [22,23], where water consists as a mixture of coherent and non-coherent domains composed of water molecules resonating at specific electromagnetic frequencies [24]. Mathematical models predict that these coherent domains could be strong devices for electromagnetic information that could organize to form a network of coherently resonating domains with greater information-storing capacity [25]. The serial dilution process can then be understood as a method of distributing coherent domains while subsequently removing all trace of the original compound. Coherent domains may be able to interact with biochemical processes to enable ultra-high dilutions to produce biological effects [26,27,28]. This phenomenon may also occur in agonist-receptor-mediated interactions, thus forming the premise for this study on angiotensin II (AngII) receptor blockers (ARBs).

Hypertension is a major health concern, contributing significantly to cardiovascular morbidity and mortality [29,30]. Hypertension is often treated pharmacologically and has contributed to a major global reduction in the incidence of cardiovascular disease [31]; however, despite this, inadequate blood pressure control persists. Due to reduced patient sensitivity to antihypertensive therapy, overmedication is relevant [32]. Patients often require at least two or more drugs for adequate blood pressure control, which inevitably increase the risk of adverse effects such as hypotension [33,34]. Therefore, the need for novel, more effective pharmaceuticals to treat hypertension is desperately needed. Sartans are a family of ARBs used to treat hypertension and other cardiovascular related diseases by targeting the renin angiotensin system (RAS) from vasoconstrictive hormones, AngII and angiotensin A (AngA). Clinical trials evaluating the efficacy of novel antihypertensives have observed reduction in systolic and diastolic blood pressure and overall lowering of blood pressure in participants in the placebo group, suggesting that the cardiovascular system may be sensitive to placebo mechanisms induced by quantum entanglement, physics, and pharmacology [1,35]. We have previously designed a series of novel sartans, based on the structure of losartan [36,37], called bisartans, which appear to be superior ARBs compared to commercially available sartans [38]. Bisartans exhibit dual antihypertensive and antiviral abilities through angiotensin type 1 receptor (AT_1_R) blockage and inhibiting entry of severe acute respiratory syndrome coronavirus 2 (SARS-CoV-2) spike protein through destabilization of the angiotensin-converting enzyme-2 (ACE2) receptor binding domain complex, and thus preventing coronavirus 2019 (COVID-19) [38,39,40,41].

The present study aimed to aid in the understanding of the placebo effect by investigating the possible existence of quantum entanglement in isolated rabbit iliac arteries and evaluating the ability of our newly synthesized bisartans to behave as novel ARBs for the potential treatment for hypertension. Arteries were incubated with known (candesartan and telmisartan) and novel (benzimidazole-*N*-biphenyl tetrazole (ACC519T), benzimidazole-*bis*-*N-N’*-biphenyl tetrazole (ACC519T(2)) and 4-butyl-N,N0-bis[[20-2Htetrazol-5-yl)biphenyl-4-yl]methyl)imidazolium bromide (BV6(K^+^)_2_) (Figure 1) ARBs synthesized by our group and nirmatrelvir (COVID-19 medication) at extremely low concentrations prepared by serial dilution in pure water in order to investigate the validity of ultra-high dilutions to provide biological effects through their ability to antagonize the AT_1_R in response to various vasopressive compounds (i.e., AngII, phenylephrine (PE) and AngA). Additionally, the ability of BV6(K^+^)_2_ to interact with ACE2 at ultra-high dilutions was also investigated, while the novel drug ACC519T, and telmisartan were also injected into conscious rabbits to determine their hemodynamic effect on mean arterial pressure (MAP).

## 2. Results

### 2.1. ACC519T Reduces MAP in a Dose-Dependent Manner

Conscious rabbits administered ACC519T showed a gradual reduction in MAP in a dose-dependent manner, as starting MAP was lowered from 114.27 ± 2.20 mmHg to 96.33 ± 1.30 mmHg at the final dose of ACC519T (Figure 2). Additionally, one rabbit required immediate euthanasia, as after administration of ACC519 10^−12^ M hypotension was observed, where MAP dropped to 30.60 mmHg and failed to return to a normal level. In contrast, the MAP of telmisartan treated rabbits remained relatively unchanged, as starting MAP was measured at 95.70 ± 3.53 mmHg and at the final dose of telmisartan was recorded at 95.90 ± 2.06 mmHg. We did notice that starting MAP differed from rabbits given ACC519T and telmisartan; however, both are considered within normal range.

### 2.2. Known and Novel ARBs Reduce AngII-Mediated Contraction at Extremely Low Concentrations

The commercially available ARBs, candesartan and telmisartan, blocked contraction in response to AngII at all doses; candesartan: 10^−6^ M (*p* < 0.0001) (Figure 3A, Table 1) to 10^−60^ M (*p* < 0.0001) (Figure 3H, Table 1) and telmisartan: 10^−6^ M (*p* < 0.0001) (Figure 3A, Table 1) to 10^−40^ M (*p* < 0.0001) (Figure 3H, Table 1). The novel ARB, ACC519T, was able to reduce AngII-induced contraction from 10^−6^ M (*p* < 0.0001) (Figure 3A, Table 1) to 10^−40^ M (*p* < 0.0001) (Figure 3F, Table 1) and at 10^−60^ M (*p* < 0.0001) (Figure 3H, Table 1). ACC519T(2) and BV6(K^+^)_2_ were able to reduce contraction in response to AngII at all doses; ACC519T(2): 10^−6^ M (*p* < 0.0001) (Figure 3A, Table 1) to 10^−40^ M (*p* < 0.0001) (Figure 3H, Table 1) and BV6(K^+^)_2_: 10^−6^ M (*p* < 0.0001) (Figure 3A, Table 1) to 10^−60^ M (*p* < 0.0001) (Figure 3H, Table 1). Importantly, nirmatrelvir had no effect on AngII contraction at any dose (Figure 3A–F, Table 1). The differences in AngII contraction responses to pre-treatment of ARBs between rabbit batches can be observed in Appendix A–L. The maximal inhibitory effect of BV6(K^+^)_2_ and candesartan, and ACC519T and telmisartan on AngII contraction are represented in Appendix A. A summary of the effects of drug pre-treatment on the inhibition of contraction responses to AngII is presented in Appendix A.

### 2.3. ACC519T Enhances Contraction While ACC519T(2) and Low-Dose BV6(K^+^)_2_, Candesartan and Telmisartan Reduce Contraction to PE

ACC519T was able to significantly enhance PE-induced contraction at 10^−6^ M (*p* < 0.0001) (Figure 4A, Table 2) and 10^−40^ M (*p* < 0.0001) (Figure 4F, Table 2). Similarly, augmented contraction to PE was observed in rings treated with candesartan at 10^−50^ M (*p* < 0.05) (Figure 4G, Table 2) and 10^−60^ M (*p* < 0.05) (Figure 4H, Table 2); however, the opposite effect was demonstrated in rings treated with candesartan 10^−12^ M (*p* < 0.01) (Figure 4B, Table 2), 10^−30^ M (*p* < 0.01) (Figure 4E, Table 2) and 10^−40^ M (*p* < 0.01) (Figure 4F, Table 2), as contraction to PE was reduced. Decreased contraction responses were also observed in rings treated with: ACC519T(2) at 10^−12^ M (*p* < 0.05) (Figure 4B, Table 2) and from doses 10^−18^ M (*p* < 0.01) (Figure 4C, Table 2) to 10^−30^ M (*p* < 0.01) (Figure 4E, Table 2); BV6(K^+^)_2_ 10^−50^ M (*p* < 0.05) (Figure 4G, Table 2); and telmisartan 10^−40^ M (*p* < 0.01) (Figure 4F, Table 2). Again, it is noted that nirmatrelvir had no effect on PE contraction at any dose (Figure 4A–F, Table 2). A summary of the effects of drug pre-treatment on the inhibition of contraction responses to PE is presented in Appendix A.

### 2.4. Novel ARBs Reduce AngA-Mediated Contraction at Extremely Low Doses

ACC519T was able to reduce AngA-induced contraction at 10^−6^ M (*p* < 0.0001) (Figure 5A, Table 3) and potently inhibited contraction from 10^−12^ M (*p* < 0.0001) (Figure 5B, Table 3) to 10^−60^ M (*p* < 0.0001) (Figure 5F, Table 3). ACC519T(2) and BV6(K^+^)_2_ were both able to insurmountable inhibit contraction to AngA at all doses; 10^−6^ M (*p* < 0.0001) (Figure 5A, Table 3) to 10^−60^ M (*p* < 0.0001) (Figure 5F, Table 3). A summary of the effects of drug pre-treatment on the inhibition of contraction responses to AngA is presented in Appendix A.

### 2.5. Novel Bisartan Does Not Affect ACE2 Activity at Extremely Low Doses

Pre-incubation of BV6(K^+^)_2_ at a 10^−6^ M was able to significantly increase ACE2 metabolism of AngII (*p* < 0.001) (Figure 6A) into Ang(1–7) (*p* < 0.001) (Figure 6B). At ultra-low concentrations of 10^−24^ M and 10^−60^ M, pre-incubation of BV6(K^+^)_2_ had no effect on AngII (Figure 6A) and Ang(1–7) (Figure 6B) concentration compared to control.

## 3. Discussion

Herein, we demonstrate the ability of commercially available ARBs, candesartan and telmisartan, and our newly synthesized imidazole-based bisartans, ACC519T, ACC519T(2) and BV6(K^+^)_2_ to significantly alter contraction responses to different vasopressor, even at ultra-low doses in two separate batches of rabbits performed at different time points. We also demonstrate that the effect of BV6(K^+^)_2_ at ultra-low doses was unable to modulate ACE2 metabolism of AngII into Ang(1–7). From these results, we hypothesize that: (1) quantum pharmacological effects and existence of quantum entanglement of substances diluted in water may contribute to the ability of ARBs to interact with certain receptors to produce a physiological effect at significantly low concentrations; (2) ARBs act like “irreversible inhibitors” in isolated tissue and may exaggerate the effects of ARBs, possibly allowing them to accumulate over time at receptors; (3) augmentation of PE response in the presence of ARB is a possible a compensatory mechanism (i.e., inactivation of the AngII receptor by ARB results in upregulation of the alpha adrenergic receptor; and (4) rabbits may be more sensitive to ARBs than humans (>1000 times), as patient daily oral dose of 10–100 mg ARB would give circulating levels of 50–500 nM, whereas 2 pg ARB per 3 kg rabbit is approximately 50 fM (this is an approximate as we are comparing oral vs. intravenous bolus).

### 3.1. Bisartan BV6(K^+^)_2_ Increases ACE2 Metabolism of AngII into Ang(1–7)

ACE2 is an important regulator of cardiovascular function by hydrolyzing AngII into Ang(1–7), reducing AngII for AT_1_R signaling [42]. Ang(1–7) can also activate the Mas receptor to produce antagonistic effects to AngII/AT_1_R signaling [43]. Increased ACE2 activity is considered therapeutically beneficial against hypertension [44], heart failure [45], and inflammation [46]. We report the ability of bisartan compound BV6(K^+^)_2_ to significantly increase ACE2 activity at 10^−6^ M as seen by the metabolism of AngII into Ang(1–7). However, in contrast to AngII-dose–response results, at ultra-diluted concentrations, BV6(K^+^)_2_ (10^−24^ M and 10^−60^ M) failed to increase enzymatic activity. Therefore, we hypothesize that the metalloenzyme ACE2 and G protein-coupled receptor AT_1_R may rely on independent ultra-high dilution–receptor interactions that are related to enzyme/receptor structure. Vibrational resonance and allosteric association have been shown to form coherent signaling pathways in binding pockets of G proteins, suggesting that specific active sites of ultra-high dilution interaction may be pivotal in proving biological responses [47,48,49].

### 3.2. Quantum Coherence, Hypersensitivity, and Receptor Entanglement as a Mechanism for Ultra-High Dilutions

This study presents experimental evidence supporting the potential of compounds to exert biological effects at concentrations where the presence of original material is considered negligible. Previous studies indicate that biological systems respond to the presence of substances at extremely low concentrations [13,50,51], suggesting that ultra-high dilutions can elicit biological are distinguishable from pure water and can provide biological responses, which is reported due to the supramolecular organization of water [52,53,54]. These studies propose that through mathematical models of quantum electrodynamics, pure liquid water exhibits particular electrodynamic behavior, supporting coherent domains and the transfer of information, which may contribute to receptor interaction through electromagnetic signal transduction during the serial dilution process [25,55,56,57]. Research suggests that electromagnetic resonance, in addition to conventional physiochemical receptor interaction, may induce changes in receptor confirmation to provide a mechanism for which ultra-diluted compounds could produce biological effects [47,58,59,60,61,62]. The concept of water “memory” is meat with heavy skepticism due to the lack of established theoretical models; however, the continuing development of quantum principles and ongoing research into the properties of water will be essential to investigating the mechanisms of this biological phenomenon.

Perceived effects of ultra-high dilutions are often attributed to the placebo effect, which suggests that cognitive and emotional changes associated with psychology contribute to observed effects [63,64]. Despite this, diverse assays, including immunologic and biochemical studies, have indicated the effects of ultra-high dilutions [65,66]. In this study, we report the ability of ultra-high dilutions of sartans to inhibit contraction in isolated rabbit vascular tissue and thus psychological changes cannot account for the observed outcomes. The placebo is commonly considered a general phenomenon, indicating that like ultra-high dilutions, there is an aspect of nature that can influence itself in a way that is not inherently understood by traditional science. Interestingly, mathematical tools derived from quantum mechanics (i.e., entanglement and interference) may provide probabilistic modelling to describe the placebo effect [5,67,68]. Quantum probability may be used to describe results that are considered paradoxical [5,69]. The hypothetical modelling proposed suggests that placebos can be associated with different outcomes even when applied to blinded clinical studies [5]. This counterintuitive model may be the consequence of quantum-like interference and may provide a framework for unexplained observation associated with the placebo effect, such as ultra-high dilutions.

Another proposed model describing the extreme sensitivity of AT_1_R to ARB involves a mechanism suggested in prions and β-amyloid protein, where a single molecule in an “incorrect” conformation triggers a cascading effect on other molecules. Similarly, ARBs may bind to AT_1_R and convert it to its “inverted state”, which in turn converts a neighboring AT_1_R into its inverted state, creating a chain reaction. Accordingly, only a single molecule of ARB may be needed to convert AT_1_R to an inverted state. Although Avogadro’s constant describes a solution diluted past 10^−24^ M as theoretically identical to the solvent and, therefore, should have no effect on biological systems, strictly speaking, Avogadro’s constant is calculated with the assumptions that equal volumes of gas, at the same pressure and temperature, contain equal number of molecules. In condensed phases, such as liquids and solids, the intermolecular interactions are significant and are far from ideal, thus liquids may not obey the law associated with Avogadro’s constant. Therefore, molecular presence of ARBs may still reside in solution diluted past 10^−24^ M; however, there must be a limit to the portion of AT_1_R, which can act in concert in this entangled manner, with the remaining AT_1_R providing normal dose–response curves, as seen at higher doses of ARBs. Here, are perhaps two types of AT_1_R in tissues: (1) the traditional functioning G protein-coupled 7 transmembrane domain receptors, which operate as dimers when transmitting the contractile response seen at higher doses of ARB, and (2) a portion of the receptors, which are identical in all respects, but are close enough in the membrane (entangled) to interact in concert [as multimers], perhaps by an amyloid-like conformation-induction mechanism, producing a response at very low levels of ARB (10^−60^ M). Our data suggests that hypersensitive entangled receptors could represent a substantial portion of the total receptors. These findings with AT_1_R could point to the existence of this type of layered (2-stage) receptor mechanism for other ligand receptors and could be a general feature of inverse agonists/inverse agonism. Most receptors act in a cooperative manner (Hill coefficient > 1), often in the form of dimers. The present findings show that receptors can also act as multimers giving a massively exaggerated response, such that a single molecule can set off a chain reaction. There may be a biological purpose to this, since human RNA is known to encode angiotensin antipeptides which are inverse agonists.

### 3.3. Inverse Agonism at AT_1_R

ARB can interact with the AT_1_R, inducing a conformational change that allows engagement of the receptor with an alternative secondary messenger that differs from the G protein associated with the contractile response [70,71,72]. This slow-to-reverse mechanism, possibly involving ligand–receptor complex internalization, may be related to prolonged desensitization akin to tachyphylaxis reported effects of ARBs [73]. In isolated smooth muscle tissue assays, the effects of peptide inverse agonists, such as sarilesin, can require hours to reverse even when the tissue is continuously washed [74]. This effect is reminiscent of irreversible (covalent) labelling of receptors [74,75], which is notably not subject to mass action laws. The time factor involved in recovery from tachyphylaxis is a result of shuttling new receptors to the membrane surface (min to hours) rather than ribosomal synthesis of new receptors (hours to days) [73]. ARBs may act as potent “tachyphylaxis-inducing agents”, potentially surpassing the longevity and effects of AngII. These irreversible properties could lead to accumulation of ligand-binding in isolated tissue experiments, which manifests as the extreme sensitivity we reported (i.e., may not be the dose per se, but rather a time-dependent accumulation). The present results may then be viewed as a consequence of irreversible effects (chemically, they are not, but functionally, for the purposes of these bioassays, they are), leading to accumulation of ligand-binding over time and ultimately resulting in extraordinary and profound sensitivity. This may be a direct outcome of the bioassay technique used in this study (i.e., long-term exposure of tissues to ARBs in cumulative tissue bath assays). However, in conscious rabbits, MAP lowering was observed after treatment with our novel sartans at doses as low as 10^−14^ M rabbit, the equivalent of ~2 pm/kg circulating levels, indicating unprecedented potency. Importantly, a dose of 10^−12^ M resulted in one rabbit requiring euthanasia due to severe hypotension. In hypertensive patients treated with sartans, circulating levels are in the micromolar range, which is many orders of magnitude higher than the effects found here in conscious rabbits.

### 3.4. ARBs and AngA

AngA was first identified in plasma from healthy and end-stage renal failure patients as a novel angiotensin-derived vasopressor [76]. The amino acid sequence of AngA differs from AngII by one amino acid (Asp is replaced with Ala) and is generated after decarboxylation of Asp in AngII by aspartate decarboxylase [76]. AngA is able to associate with AT_1_R; although with a lower affinity compared to AngII [76]. In isolated perfused rat kidneys, AngA was able to induce a small vasoconstrictive effect, suggesting that AngA is a less potent and partial agonist at the AT_1_R [76]. Moreover, AngA pressor and renal vasoconstrictive abilities were almost abolished in mice with AT_1_R(a) deletion [77]. In contrast, administration of cumulative doses of AngA caused increased blood pressure, with comparable magnitude to AngII, which was only partially blocked by AT_1_R inhibitor losartan [78]. Our results show that the ability of ACC519T, BV6(K^+^)_2_ and candesartan were also able to insurmountably inhibit AngA-mediated contraction at low doses, suggesting that these inhibitors may reduce effects of the alternative RAS peptide which could be involved in certain pathologies.

### 3.5. ARBs, PE and Agonism and Compensatory Mechanisms

There is limited literature available on the ability of ARBs to alter contraction in response to PE. A study conducted on rat femoral arteries revealed that a 30 min incubation with telmisartan [30 µM] but not losartan [30 µM] was able to significantly reduce contraction to PE dose–response effect [79]. When studies of ARB effects were studied using tissues contracted with an alternative agonist, namely the α1 adrenergic agonist PE, the effects of ARBs were generally lowering of the contractile response. However, we have previously shown that at 10^−6^ M, ACC519T augmented the response to PE, roughly doubling the contractile response. It appears unlikely that this counterintuitive finding results from the release of adrenaline from limited tissue stores in response to ARB but may be the result of biased or compensatory agonism.

The term “biased agonism” was developed to explain functional selectivity of G protein-coupled receptors in signaling pathways, such as those seen with ARBs resulting in inverted AT_1_R function. More complex properties of receptors, wherein several different ligands can act at different sites on one or more receptors to produce a “summation” effect can also occur, presumably as a means for fine-tuning the physiological response. Thus, a second ligand (like an ARB) could theoretically augment or attenuate the response to another primary agonist (e.g., PE), or even produce the reverse (inverse) response.

Considerations, as previously described, may partly explain the anomalies observed in these studies, such as the reverse dose–response effect of ACC519T(2) in AngII assays (e.g., binding to two competing sites with opposite regulatory effects). Likewise, the mechanism by which ACC519T augments the PE response is unknown. However, since ACC519T(2) is structurally unrelated to PE, a possibility exists that the exacerbated response could be the product of two different binding sites located on the α1 adrenergic receptor, leading to a combined effect. For example, the α1 receptor may conceivably support a secondary binding site for AngII, which permits this peptide to attenuate the α1 receptor and vice versa. Such a mechanism could make sense as a means of preventing cardiovascular overkill by two potent vasoconstrictors (e.g., AngII, AngA, noradrenaline and adrenaline) during simultaneous release. As a corollary to this logic, ARBs may also bind to this modulatory AngII binding site, causing up or downregulation of the AngII receptor.

The mechanism by which ACC519T augments the PE response is unknown. Another explanation is that there is “crosstalk” between the AT_1_R and α1 adrenergic receptor, and inhibition of the AT_1_R results in a compensatory up-sensitization of the α receptor or its coupling/second messenger mechanism. Thus, deactivation of one receptor leads to upregulation of the other. This compensatory mechanism could derive from direct allosteric effects between receptors upregulating expression against each other in the membrane or through interactions between their response transmission mechanisms (e.g., G proteins, calcium ions). It appears that receptors can be complex gatekeepers, exhibiting not just biased agonism but, more broadly, “compensatory or summative” effects of different ligands.

### 3.6. ARBs as a COVID-19 Treatment

The beneficial role of RAS inhibitors, such as ARBs, in the treatment of COVID-19 have clinically been investigated due to the association between SARS-CoV-2 infection, RAS dysfunction and cardiovascular complications [80,81,82,83]. The protective effect of telmisartan to potentially reduce morbidity and mortality in SARS-CoV-2 patients [84] and candesartan to ameliorate the COVID-19 cytokine storm [85] has been reported. In this study, we synthesized benzimidazole biphenyl tetrazole analogues with one and two biphenyl arms built on the same benzimidazole scaffold as candesartan and telmisartan. Furthermore, ARBs exert protective actions not only as vasodilators but also as modulators of gene expression, cell growth, autoimmune disease (e.g., rheumatoid arthritis and multiple sclerosis) and inflammation to provide a promising strategy for COVID-19 [86,87]. The protective effect of telmisartan to potentially reduce morbidity and mortality in SARS-CoV-2 patients prompted the synthesis and evaluation of benzimidazole biphenyl sartans, which bear the same scaffold as candesartan, telmisartan and azilsartan. Bisartans exhibit dual antihypertensive and antiviral abilities through AT_1_R blockage and inhibiting entry of SARS-CoV-2 spike protein through ACE2 or destabilization of the ACE2/receptor binding domain complex, preventing COVID-19 infection. Moreover, docking studies have shown bisartans to be superior blockers compared to commercially available sartans, highlighting the importance of the tetrazole moiety. The computational and clinical studies of ARBs favor their development as strong competitors of the newly introduced Paxlovid, the antiviral drug Pfizer developed to treat COVID-19 patients [38,80,88,89,90]. Importantly, there is evidence that the COVID-19 medication Paxlovid, of which nirmatrelvir is the main active ingredient (Pfizer, New York, NY, USA), may cause unwanted drug–drug interactions with medications commonly prescribed to patients with pre-existing cardiovascular diseases [91,92]. This, coupled with our results showing the inability of nirmatrelvir to modulate AngII- and AngA-mediated contraction even at clinically relevant doses, demonstrates the importance of investigating potential cardiovascular-protective pharmaceuticals to be used in conjunction with Paxlovid is essential.

## 4. Materials and Methods

### 4.1. Chemicals, Pharmaceuticals, and Peptides

Ang(1–7) was purchased from AdooQ Bioscience (Cat#A14862) (Irvine, CA, USA); acetonitrile (Cat#AJA2315) and zinc acetate (Cat#CM2750) were purchased from Ajax Chemicals (Taren Point, NSW, Australia); nirmatrelvir/PF-07321332 (Cat#AOB14800) was purchased from Aobious (Gloucester, MA, USA); human AngII sequence DRVYIHPF (Cat#51480) and AngA sequence ARVYIHPF (Cat# 100365) were purchased from Mimitopes (Mulgrave, VIC, Australia); rhACE2 (Cat#933-ZN-010) was purchased from R&D systems (Minneapolis, MN, USA); telmisartan (Cat#S1738) was purchased from Selleck Chemicals (Houston, TX, USA); and CaCl_2_ (Cat#C1016), candesartan (Cat#SML0245), formic acid (Cat#543804), glucose (Cat#G7021), KCl (Cat#P9541), KH_2_PO_4_ (Cat#P0662), MgSO_4_·7H_2_O (Cat#230391), NaHCO_3_ (Cat#S5761), NaCl (Cta#S9888) and (R)-(−)-phenylephrine hydrochloride (Cat#P6126) were purchased from Sigma Aldrich (St. Louis, MO, USA).

### 4.2. Synthesis of Experimental ARBs

The synthesis of ACC519T and ACC519T(2) involves a two-step procedure, as previously described [37]. Briefly, mono- or *bis*-alkylation of benzimidazole using alkylating reagent 4-(bromomethyl)-[1,1-biphenyl ]-2-(trityl) tetrazole using a 1:1 or 1:2 molar ratio, respectively, followed by removal of trityl groups [37]. Due to the similar chemical structure shared between ACC519T, ACC519T(2) and telmisartan, and BV6(K^+^)_2_ and candesartan, these commercially available and commonly prescribed pharmaceuticals were used in this experiment as positive controls with known effects to validate the effectiveness of our newly synthesized ARBs.

### 4.3. Animal Model, Care, and Ethics Approval

Male New Zealand White rabbits (*n* = 10) at 8 weeks (wk) of age were purchased in two separate batches from Flinders City University (Adelaide, SA, Australia) and were housed at Victoria University, Werribee Campus Animal Facilities, VIC Australia. Animals were given a 7-d acclimation period and were housed in pairs until 10 wk of age before reaching sexual maturity. Animals were aged to 16 wk and were kept on a 12-h day/night cycle and maintained at a constant temperature of 21 °C and relative humidity level between 40 and 70%. Rabbits were fed normal chow diet pellets (Speciality Feeds, Glen Forrest, WA, Australia) and food and water were supplied ad libitum. The first batch of rabbits (*n* = 6) were used for a preliminary pilot to determine traditional Emax and evaluate the ability of experimental ARBs to alter contraction responses in iliac artery rings. The second batch of rabbits (*n* = 4) was used to confirm the ability of ARBs to reduce contraction at low doses. All experimental procedures were conducted in accordance with the National Health and Medical Research Council ‘Australia Code of Practice for the Care and Use of Animals for Scientific Purposes’ (8th edition), 2013; https://www.nhmrc.gov.au/about-us/publications/australian-code-care-and-use-animals-scientific-purposes (accessed on 10 November 2023)) and was approved by the Victoria University Animal Ethics Committee (VUAEC#17/013).

Rabbits were chosen for this study as they share closer phylogenetic resemblance to humans, and pioneering studies derived from rabbit experiments support their use as a more reliable model for studying cardiovascular physiology, mechanisms, and pathology than laboratory rodents (e.g., mice, rats, and guinea pigs), which have limited translational impact [93]. Importantly, rabbits and humans elicit an almost identical vascular response to AngII in conduit arteries. An often-ignored problem inherent to the use of rodent models is the existence of two AT_1_R isoforms, namely AT_1_R subtype A and B [94], the latter of which is predominantly responsible for mediating AngII vasoconstriction responses [95]. However, we [96] and others [95,97] have shown the inability or reduced capacity of AngII to stimulate contraction responses in rodent blood vessels, including the thoracic and abdominal aorta, and carotid and brachiocephalic arteries. This may be explained by reduced sensitivity to AngII, desensitization of AT1R subtypes subsequent to successive AngII exposure and poor expression profiles in blood vessels [98].

### 4.4. Rabbit MAP Measuring

Indirectly measuring blood pressure by Doppler ultrasound or non-invasive oscillometric monitors can provide unreliable and greatly varying results in rabbits. Therefore, directly measuring MAP via central auricular artery catheterization has become a standard practice. Rabbits were towel wrapped to reduce stress during the procedure and the hair on the left and right ears was removed for greater visualization of the auricular vasculature. To reduce discomfort during the procedure, Numit topical cream (lidocaine 2.5% *w*/*w* and prilocaine 2.5% *w*/*w*) was liberally applied and allowed to absorb for 15–20 min. Heparin coated IV catheters were introduced into the left lateral marginal auricular vein (24 G) and right central auricular artery (22 G) and were securely held in place with sensitive adhesive tape. ACC519T or telmisartan were administered via the marginal auricular vein and alterations in MAP were recorded and compared by connecting the arterial catheter to a pressure transducer (BP1, Zultek Engineering, Melbourne, VIC, Australia) to a PowerLab 2/20 (ADInstrument, Sydney, NSW, Australia). Once catheters were placed into the ear, rabbits were allowed a 5 min rest period for MAP stabilization. ACC519T and telmisartan were then administered in a dose–response effect [10^−14^ M to 10^−10^ M], with each dose being injected after 5 min. At the end of the procedure, catheters were removed from rabbit ears and rabbits were monitored for 2 h.

### 4.5. Anaesthesia and Humane Dispatch Protocol

Rabbits were first sedated using a subcutaneous injection of medetomidine (0.25 mg/kg) at the ‘scruff’ or base of the neck to reduce stress and anesthetized using the inhalant isoflurane (4%). When loss of corneal and palpebral pain reflex was observed, an incision was made at the lower abdomen and the subcutaneous tissue and muscles were dissected to expose the inferior vena cava. Rabbits were humanely dispatched by inferior vena cava exsanguination and death was signified by dissection of the diaphragm. A T-tube was introduced distal to the aortic arch to allow adequate flushing of the aorta, aortic bifurcation, and iliac arteries with cold (4 °C) oxygenated Krebs–Henseleit (Krebs) (118 mM, NaCl; 4.7 mM KCl; 1.2 mM MgSO_4_·7H_2_O; 1.2 mM KH_2_PO_4_; 25 mM NaHCO_3_; 11.7 mM glucose; and 1.25 mM CaCl_2_) (pH: 7.4). The left and right iliac arteries were retrieved from each animal and, under a light microscope, were cleaned of connective and adipose tissue and cut into 2 mm rings for isometric tension myography studies. A 2 week ‘wash out’ period was allowed after administration of ARBs to live rabbits to ensure ex vivo isometric tension results were not contaminated by ACC519T or telmisartan.

### 4.6. Drug Incubations and Isometric Tension Myography Studies

Isometric tension myography is a gold-standard functional test used to determine the pharmacodynamic effects that a pharmaceutical has on blood vessel function. Importantly, these studies allow a real-time trace to determine the effect that a drug has on the ability of blood vessels to relax or contract to different vasoactive peptides, providing an ex vivo understanding of isolated and potentially systemic effects that a drug may have on blood vessel function. Iliac artery rings were immediately and sequentially placed into adjacent organ baths (OB16, Zultek Engineering, Melbourne, VIC, Australia) filled with 5 mL of Krebs and acclimatized for 30 min. To replicate a physiologically relevant environment, baths were maintained at 37 °C and continuously bubbled with carbogen (95% O_2_/5% CO_2_). After acclimation, rings were mounted between two metal organ hooks attached to force displacement transducers, stretched to 0.5 g, and equilibrated for a further 30 min. Rings were then refreshed, re-stretched, and again equilibrated for 30 min before drug incubations. To investigate the antihypertensive abilities of novel and commercially available ARBs, rings severing as control groups were not incubated with any drug and left to rest for 10 min, while in ARB experimental groups rings, were incubated with ACC519T, ACC519T(2), BV6(K^+^)_2_, candesartan, telmisartan or nirmatrelvir for final concentrations of 10^−5^ M, 10^−6^ M, 10^−7^ M, 10^−8^ M, 10^−9^ M, 10^−10^ M, 10^−11^ M, 10^−12^ M, 10^−13^ M, 10^−14^ M and 10^−15^ M for 10 min in batch 1 rabbits as trial experiments. Across both batches of rabbits, ARBs were also incubated at 10^−18^ M, 10^−24^ M, 10^−30^ M, 10^−40^ M, 10^−50^ M, and 10^−60^ M for 10 min. An AngII (10^−11^ M–10^−5^ M), PE (10^−9^ M–10^−5^ M) or AngA (10^−11^ M–10^−5^ M) dose–response study was then performed to determine the ability of drugs to alter contraction (graphs depicting incubations with 10^−5^ M, 10^−7^ M–10^−11^ M and 10^−13^ M–10^−15^ M followed by AngII dose–response effect are represented in Appendix A–I). Following the completion of dose–response studies, rings were refreshed, allowed to return to baseline tension, and contracted with high potassium physiological solution (KPSS) [125 mM] (125 mM/L KCl; 1.2 mM/L MgSO_4_·7H_2_O; 1.2 mM/L KH_2_PO_4_; 25 mM/L NaHCO_3_; and 11.7 mM/L glucose; and 1.25 mM CaCl_2_) (pH: 7.4) to determine maximal standard contraction responses. Due to some doses being performed in both batches of rabbits and others only in the first or second batch, differences in the number of iliac rings retriable from each rabbit and rings that did not respond to KPSS being omitted from the study, *n* = 3 as observed across both rabbit batches. Preparation of ARB concentration occurred through serial dilution in which 1:1000 dilutions were performed in individual plastic Eppendorf tubes until the desired concentration was reached. Solutions were vortexed between each concurrent dilution.

### 4.7. Bisartan Incubation and ACE2 Activity

The ability of BV6(K^+^)_2_ to interact with ACE2 at ultra-high dilutions was investigated by incubation of BV6(K^+^)_2_ in a total volume of 2 mL containing a buffer solution (50 mM/L HEPES; 100 mM/L NaCl; and 10 μM/L Zinc acetate) (pH: 7.0) along with 50 ng/mL of protein. Sample without (control) or with BV6(K^+^)_2_ performed in triplicate at final concentrations of 10^−6^ M, 10^−24^ M, and 10^−30^ M were pre-incubated with rhACE2 at 37 °C for 10 min prior to addition of 100 μM AngII. As soon as the substrate was added, samples were vortexed and incubated for 60 min at 37 °C. The reaction was stopped by heating the samples to 80 °C for 5 min to denature the enzyme, after which samples were placed on ice. To measure the effects of BV6(K^+^)_2_ on ACE2 metabolism, AngII and the product Ang(1–7) were separated and quantified on a series LC-2030 HPLC (Shimadzu Corporation, Kyoto, Japan) using an Agilent 300C C18 reverse-phase column (2.1 × 150 mm, 5 μM) and a UV-detector set at 220 nM (Agilent Technologies, Santa Clara, CA, USA). All separations were conducted at room temperature, with a flow rate of 0.4 mL/min. Mobile phases consisted of A; HPLC grade MilliQ water + 0.1% formic acid, and B; acetonitrile + 0.1% formic. A solvent of gradient of 10–70% B, over 9 min, 70–10% B, over 2 min, and 4 min on final concentration with a 10 min re-equilibration phase was used. Identification of peptides and quantification was based upon the elution time of synthetic standards and known amounts of AngII and Ang(1–7). The mean absorbance of replicates between each samples was calculated, and the standard curve was used to interpolate values of both peptides.

### 4.8. Statistical Analysis

GraphPad prism (version 9.5.1) was utilized for statistical analysis of isometric tension, MAP data, and ACE2 activity. A one-way ANOVA followed by Dunnett’s (ACE2) or Sidak’s post hoc was utilized to determine significance in ACE2 activity and MAP studies, respectively. A two-way ANOVA followed by Sidak’s post hoc was used to determine significance in isometric tension myography results. The significant *p*-value was set at *p* < 0.05, and all data are represented as the mean ± standard error of mean (SEM).

## 5. Conclusions

Herein, we demonstrate the ability of extraordinarily low doses of both our novel and commonly prescribed ARBs to inhibit AngII and AngA contraction responses and augment PE contraction responses in isolated rabbit iliac arteries. Incredibly, we also show that a low dose of the ARB ACC519T is able to lower MAP in conscious rabbits. Receptor responses can be extremely complex due to electromagnetic activity, “cross interactions” among ligands, receptors, and intracellular signaling mechanisms. We speculate the mechanism of ultra-low dilutions to be aligned with quantum effects of ARBs diluted in pure water where their antagonistic effects may be retained through quantum electromagnetic effects to produce receptor interaction, which may aid in explaining the placebo effect that is seen in randomized controlled trials of different drugs. Moreover, the ability of our newly synthesized bisartans and commercially available ARBs to have an effect at extremely low concentrations challenges the current dosage given to hypertensive patients. Additionally, augmentation of PE responses in the presence of ACC519T may be due to receptor “cross talk” wherein deactivation of the AngII receptor is offset by upregulation of the α adrenergic receptor response mechanism, possibly via interlinked second messenger systems. Likewise, the reversal of the dose–response effect of ACC519T(2), in which low doses (10^−40^ M) block AngII but high doses (10^−6^ M) do not, contrasts with the ACC519T(2) blockade of AngA at all doses (which is similar to ACC519T and BV6(K^+^)_2_) and suggests a different receptor mechanism for AngA. Future in silico and molecular dynamic docking studies may be useful in investigating the different mechanism of ARB receptor interaction. It is essential that further investigation into the underlying mechanism of our results is performed and that other research groups can replicate our findings in order to validate the effects of ultra-high dilutions on various biological systems.

## Figures and Tables

**Figure 1 ijms-24-17559-f001:**
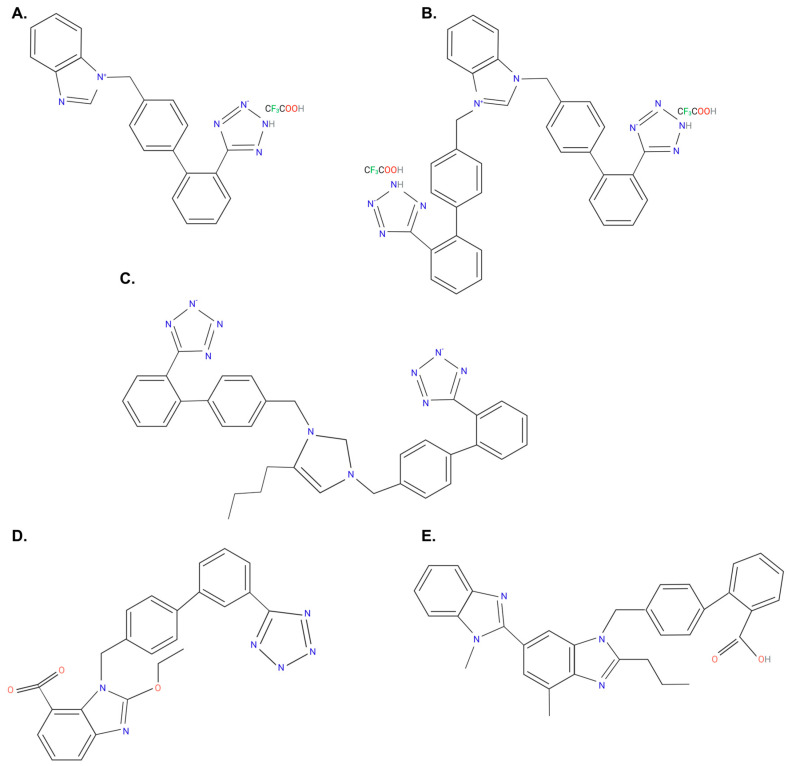
Chemical structure of (**A**) ACC519T, (**B**) ACC519T(2), (**C**) BV6(K^+^)_2_, (**D**) candesartan, and (**E**) telmisartan. Abbreviations: ACC519T, benzimidazole-*N*-biphenyl tetrazole; ACC519T(2), benzimidazole-bis-*N*,*N*′-biphenyl tetrazole; BV6(K^+^)_2_, 4-butyl-N,N0-bis[[20-2Htetrazol-5-yl)biphenyl-4-yl]methyl)imidazolium bromide. Black: carbon; green: fluorine; gray: hydrogen; blue: nitrogen and red: oxygen. Figure created using Biorender.com (access on 9 October 2023).

**Figure 2 ijms-24-17559-f002:**
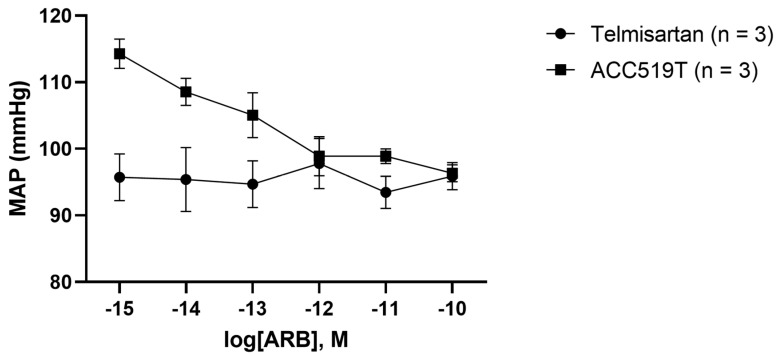
Changes in MAP of rabbits administered novel and commercially available ARBs. Dose–response effect of either ACC519T or telmisartan in rabbits with MAP recorded. Treatment telmisartan was unable to invoke a reduction in MAP pressure at this dose range, which remained relatively unchanged (mean ± SEM). However, treatment with ACC519T resulted in a gradual reduction in MAP in a dose-dependent manner (mean ± SEM). Abbreviations: ACC519T or telmisartan via ear vein catheterization. Abbreviations: ACC519T, benzimidazole-*N*-biphenyl tetrazole, MAP, mean arterial pressure; SEM, standard error of mean.

**Figure 3 ijms-24-17559-f003:**
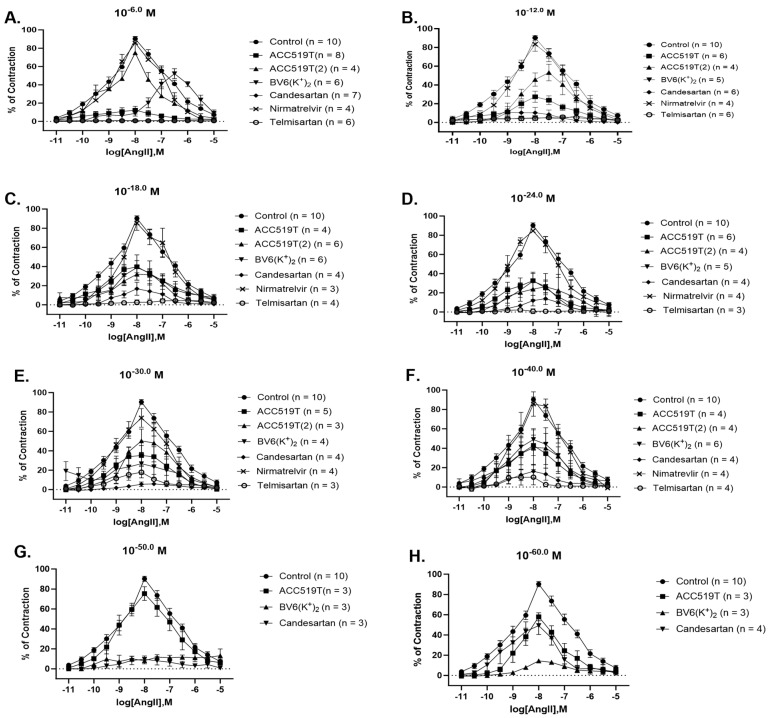
Contraction responses to AngII dose–response effect in rabbit iliac arteries left to rest (control) or pre-treated with various doses of commercial and experimental ARBs. Contraction responses of rabbit iliac to AngII dose–response effect after pre-treatment with (**A**) 10^−6^ M, (**B**) 10^−12^ M, (**C**) 10^−18^ M, (**D**) 10^−24^ M, (**E**) 10^−30^ M, (**F**)10^−40^ M, (**G**) 10^−50^ M, and (**H**) 10^−60^ M dose of ACC519T, ACC519T(2), BV6(K^+^)_2_, candesartan, nirmatrelvir and telmisartan (mean ± SEM) (significance shown in Table 1). Abbreviations: ACC519T, benzilimidazole-*N*-biphenyl tetrazole; ACC519T(2), benzimidazole-bis-*N,N*′-biphenyl tetrazole; AngII, angiotensin II; ARBs, angiotensin receptor blockers; BV6(K^+^)_2_, 4-butyl-N,N0-bis[[20-2Htetrazol-5-yl)biphenyl-4-yl]methyl)imidazolium bromide; SEM, standard error of mean.

**Figure 4 ijms-24-17559-f004:**
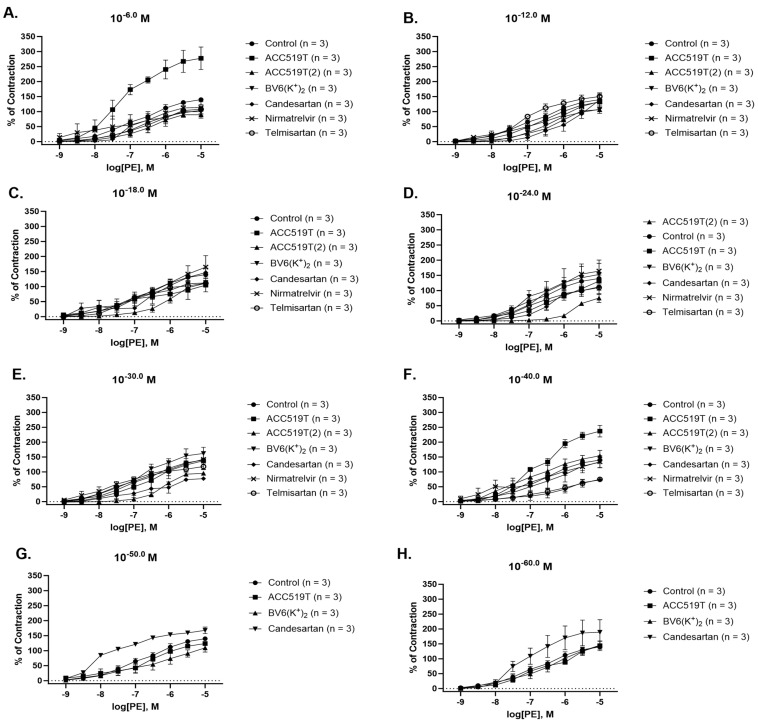
Contraction responses to PE dose–response effect in rabbit iliac arteries left to rest (control) or pre-treated with various doses of experimental drugs. Contraction responses of rabbit iliac to PE dose–response effect after ore-treatment with (**A**) 10^−6^ M, (**B**) 10^−12^ M, (**C**) 10^−18^ M, (**D**) 10^−24^ M, (**E**) 10^−30^ M, (**F**) 10^−40^ M, (**G**) 10^−50^ M, and (**H**) 10^−60^ M dose of ACC519T, ACC519T(2), BV6(K^+^)_2_, candesartan, nirmatrelvir and telmisartan (mean ± SEM) (significance shown in Table 2). Abbreviations: ACC519T, benzilimidazole-*N*-biphenyl tetrazole; ACC519T(2), benzimidazole-*bis*-*N*,*N*′-biphenyl tetrazole; ARBs, angiotensin receptor blockers; BV6(K^+^)_2_, 4-butyl-N,N0-bis[[20-2Htetrazol-5-yl)b-phenyl-4-yl]methyl)imidazolium bromide; PE, phenylephrine; SEM, standard error of mean.

**Figure 5 ijms-24-17559-f005:**
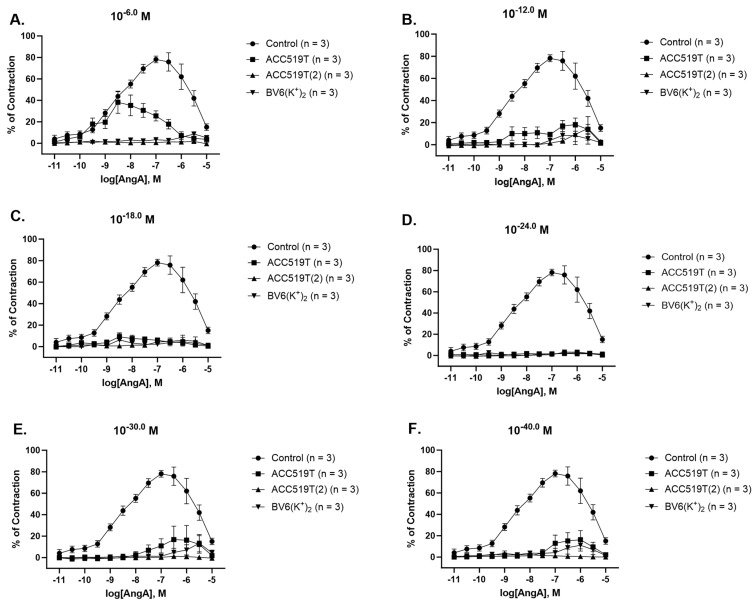
Contraction responses to AngA dose–response effect in rabbit iliac arteries left to rest (control)or pre-treated with various doses of ARBs. Contraction responses of control and ACC519T, ACC519T(2) and BV6(K^+^)_2_ treated ring at (**A**) 10^−6^ _M_, (**B**) 10^−12^ _M_, (**C**) 10^−18^ _M_, (**D**) 10^−24^ _M_, (**E**) 10^−30^ _M_ and (**F**) 10^−40^ _M_ after AngA dose–response effect (mean ± SEM) (significance shown in Table 3). Abbreviations: ACC519T, benzilimidazole-*N*-biphenyl tetrazole; ACC519T(2), benzimidazole-*bis*-*N*,*N*′-biphenyl tetrazole; AngA, angiotensin A; ARBs, angiotensin receptor blockers; BV6(K^+^)_2_, 4-butyl-N,N0-bis[[20-2Htetrazol-5-yl)biphenyl-4-yl]methyl)imidazolium bromide; SEM, standard error of mean.

**Figure 6 ijms-24-17559-f006:**
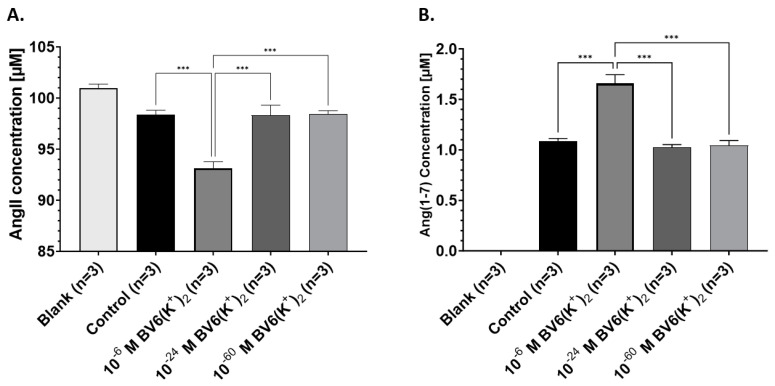
rhACE2 activity response to pre-incubation of BV6(K^+^)_2_ at various doses performed in triplicate (*n* = 3). Retrieved concentration [μM] of (**A**) AngII and (**B**) Ang(1–7) as a result of AngII metabolism by rhACE2 with and without the presence of 10^−6^ M, 10^−24^ M, and 10^−30^ M BV6(K^+^)_2_ (mean ± SEM). *** *p* < 0.001 (one-way ANOVA). Abbreviations: BV6(K^+^)_2_, 4-butyl-N,N0-bis[[20-2Htetrazol-5-yl)biphenyl-4-yl]methyl)imidazolium bromide; rhAEC2, human recombinant angiotensin-converting enzyme-2; SEM, standard error of mean. Figure created using Biorender.com.

**Table 1 ijms-24-17559-t001:** Significant differences in contraction responses to AngII dose–response effect after pre-treatment with various doses of drugs. Abbreviations: ACC519T, benzilimidazole-*N*-biphenyl tetrazole; ACC519T(2), benzimidazole-bis-*N,N*′-biphenyl tetrazole; BV6(K^+^)_2_, 4-butyl-N,N0-bis[[20-2Htetrazol-5-yl)biphenyl-4-yl]methyl)imidazolium bromide; ns, no significance.

**log[AngII], M**	**ACC519T** **10^−6^ M vs. Control**	**ACC519T(2)** **10^−6^ M vs. Control**	**BV6(K^+^)_2_** **10^−6^ M vs. Control**	**Candesartan** **10^−6^ M vs. Control**	**Nirmatrelvir** **10^−6^ M vs. Control**	**Telmisartan** **10^−6^ M vs. Control**
**−11.0**	ns	ns	ns	ns	ns	ns
**−10.5**	ns	ns	ns	ns	ns	ns
**−10.0**	*p* < 0.01	ns	*p* < 0.01	*p* < 0.001	ns	*p* < 0.001
**−9.5**	*p* < 0.0001	ns	*p* < 0.0001	*p* < 0.0001	ns	*p* < 0.0001
**−9.0**	*p* < 0.0001	ns	*p* < 0.0001	*p* < 0.0001	ns	*p* < 0.0001
**−8.5**	*p* < 0.0001	ns	*p* < 0.0001	*p* < 0.0001	ns	*p* < 0.0001
**−8.0**	*p* < 0.0001	*p* < 0.05	*p* < 0.0001	*p* < 0.0001	ns	*p* < 0.0001
**−7.5**	*p* < 0.0001	*p* < 0.0001	*p* < 0.0001	*p* < 0.0001	ns	*p* < 0.0001
**−7.0**	*p* < 0.0001	*p* < 0.0001	*p* < 0.01	*p* < 0.0001	ns	*p* < 0.0001
**−6.5**	*p* < 0.0001	*p* < 0.001	ns	*p* < 0.0001	ns	*p* < 0.0001
**−6.0**	*p* < 0.0001	ns	*p* < 0.001	*p* < 0.0001	ns	*p* < 0.0001
**−5.5**	*p* < 0.05	ns	ns	*p* < 0.05	ns	*p* < 0.05
**−5.0**	ns	ns	ns	ns	ns	ns
**log[AngII], M**	**ACC519T** **10^−12^ M vs. Control**	**ACC519T(2)** **10^−12^ M vs. Control**	**BV6(K^+^)_2_** **10^−12^ M vs. Control**	**Candesartan** **10^−12^ M vs. Control**	**Nirmatrelvir** **10^−12^ M vs. Control**	**Telmisartan** **10^−12^ M vs. Control**
**−11.0**	ns	ns	ns	ns	ns	ns
**−10.5**	ns	ns	ns	ns	ns	ns
**−10.0**	ns	ns	ns	ns	ns	ns
**−9.5**	*p* < 0.05	*p* < 0.05	*p* < 0.01	*p* < 0.01	ns	*p* < 0.01
**−9.0**	*p* < 0.001	*p* < 0.01	*p* < 0.001	*p* < 0.001	ns	*p* < 0.0001
**−8.5**	*p* < 0.0001	*p* < 0.01	*p* < 0.0001	*p* < 0.0001	ns	*p* < 0.0001
**−8.0**	*p* < 0.0001	*p* < 0.0001	*p* < 0.0001	*p* < 0.0001	ns	*p* < 0.0001
**−7.5**	*p* < 0.0001	ns	*p* < 0.0001	*p* < 0.0001	ns	*p* < 0.0001
**−7.0**	*p* < 0.0001	ns	ns	*p* < 0.0001	ns	*p* < 0.0001
**−6.5**	*p* < 0.01	ns	ns	*p* < 0.0001	ns	*p* < 0.0001
**−6.0**	ns	ns	ns	ns	ns	ns
**−5.5**	ns	ns	ns	ns	ns	ns
**−5.0**	ns	ns	ns	ns	ns	ns
**log[AngII], M**	**ACC519T** **10^−18^ M vs. Control**	**ACC519T(2)** **10^−18^ M vs. Control**	**BV6(K^+^)_2_** **10^−18^ M vs. Control**	**Candesartan** **10^−18^ M vs. Control**	**Nirmatrelvir** **10^−18^ M vs. Control**	**Telmisartan** **10^−18^ M vs. Control**
**−11.0**	ns	ns	ns	ns	ns	ns
**−10.5**	ns	ns	ns	ns	ns	ns
**−10.0**	ns	ns	ns	*p* < 0.05	ns	ns
**−9.5**	ns	*p* < 0.01	*p* < 0.05	*p* < 0.001	ns	*p* < 0.001
**−9.0**	*p* < 0.05	*p* < 0.0001	*p* < 0.0001	*p* < 0.0001	ns	*p* < 0.0001
**−8.5**	*p* < 0.01	*p* < 0.0001	*p* < 0.0001	*p* < 0.0001	ns	*p* < 0.0001
**−8.0**	*p* < 0.0001	*p* < 0.0001	*p* < 0.0001	*p* < 0.0001	ns	*p* < 0.0001
**−7.5**	*p* < 0.0001	*p* < 0.0001	*p* < 0.0001	*p* < 0.0001	ns	*p* < 0.0001
**−7.0**	*p* < 0.001	*p* < 0.0001	*p* < 0.0001	*p* < 0.0001	ns	*p* < 0.0001
**−6.5**	*p* < 0.01	*p* < 0.01	*p* < 0.0001	*p* < 0.0001	ns	*p* < 0.0001
**−6.0**	ns	ns	*p* < 0.05	*p* < 0.05	ns	ns
**−5.5**	ns	ns	ns	ns	ns	ns
**−5.0**	ns	ns	ns	ns	ns	ns
**log[AngII], M**	**ACC519T** **10^−24^ M vs. Control**	**ACC519T(2)** **10^−24^ M vs. Control**	**BV6(K^+^)_2_** **10^−24^ M vs. Control**	**Candesartan** **10^−24^ M vs. Control**	**Nirmatrelvir** **10^−24^ M vs. Control**	**Telmisartan** **10^−24^ M vs. Control**
**−11.0**	ns	ns	ns	ns	ns	ns
**−10.5**	ns	ns	ns	ns	ns	ns
**−10.0**	ns	ns	ns	ns	ns	ns
**−9.5**	ns	*p* < 0.01	*p* < 0.05	*p* < 0.01	ns	*p* < 0.01
**−9.0**	*p* < 0.05	*p* < 0.01	*p* < 0.01	*p* < 0.0001	ns	*p* < 0.0001
**−8.5**	*p* < 0.0001	*p* < 0.0001	*p* < 0.0001	*p* < 0.0001	ns	*p* < 0.0001
**−8.0**	*p* < 0.0001	*p* < 0.0001	*p* < 0.0001	*p* < 0.0001	ns	*p* < 0.0001
**−7.5**	*p* < 0.0001	*p* < 0.0001	*p* < 0.0001	*p* < 0.0001	ns	*p* < 0.0001
**−7.0**	*p* < 0.0001	*p* < 0.001	*p* < 0.0001	*p* < 0.0001	ns	*p* < 0.0001
**−6.5**	*p* < 0.0001	*p* < 0.05	*p* < 0.0001	*p* < 0.0001	ns	*p* < 0.0001
**−6.0**	ns	ns	*p* < 0.01	ns	ns	ns
**−5.5**	ns	ns	ns	ns	ns	ns
**−5.0**	ns	ns	ns	ns	ns	ns
**log[AngII], M**	**ACC519T** **10^−30^ M vs. Control**	**ACC519T(2)** **10^−30^ M vs. Control**	**BV6(K^+^)_2_** **10^−30^ M vs. Control**	**Candesartan** **10^−30^ M vs. Control**	**Nirmatrelvir** **10^−30^ M vs. Control**	**Telmisartan** **10^−30^ M vs. Control**
**−11.0**	ns	ns	ns	ns	ns	ns
**−10.5**	ns	ns	ns	ns	ns	ns
**−10.0**	ns	ns	ns	ns	ns	ns
**−9.5**	ns	ns	ns	*p* < 0.05	ns	ns
**−9.0**	ns	*p* < 0.01	ns	*p* < 0.001	ns	*p* < 0.05
**−8.5**	ns	*p* < 0.05	ns	*p* < 0.0001	ns	*p* < 0.001
**−8.0**	*p* < 0.001	*p* < 0.0001	*p* < 0.0001	*p* < 0.0001	ns	*p* < 0.0001
**−7.5**	*p* < 0.05	*p* < 0.001	*p* < 0.01	*p* < 0.0001	ns	*p* < 0.0001
**−7.0**	*p* < 0.05	*p* < 0.05	*p* < 0.05	*p* < 0.0001	ns	*p* < 0.0001
**−6.5**	*p* < 0.05	ns	ns	*p* < 0.001	ns	*p* < 0.01
**−6.0**	ns	ns	ns	ns	ns	ns
**−5.5**	ns	ns	ns	ns	ns	ns
**−5.0**	ns	ns	ns	ns	ns	ns
**log[AngII], M**	**ACC519T** **10^−40^ M vs. Control**	**ACC519T(2)** **10^−40^ M vs. Control**	**BV6(K^+^)_2_** **10^−40^ M vs. Control**	**Candesartan** **10^−40^ M vs. Control**	**Nirmatrelvir** **10^−40^ M vs. Control**	**Telmisartan** **10^−40^ M vs. Control**
**−11.0**	ns	ns	ns	ns	ns	ns
**−10.5**	ns	ns	ns	ns	ns	ns
**−10.0**	ns	ns	ns	ns	ns	ns
**−9.5**	ns	ns	ns	*p* < 0.01	ns	*p* < 0.05
**−9.0**	ns	ns	ns	*p* < 0.001	ns	*p* < 0.01
**−8.5**	*p* < 0.05	*p* < 0.05	ns	*p* < 0.0001	ns	*p* < 0.0001
**−8.0**	*p* < 0.0001	*p* < 0.0001	*p* < 0.0001	*p* < 0.0001	ns	*p* < 0.0001
**−7.5**	*p* < 0.0001	*p* < 0.001	*p* < 0.001	*p* < 0.0001	ns	*p* < 0.0001
**−7.0**	*p* < 0.01	*p* < 0.05	*p* < 0.05	*p* < 0.0001	ns	*p* < 0.0001
**−6.5**	*p* < 0.01	*p* < 0.05	*p* < 0.01	*p* < 0.001	ns	*p* < 0.001
**−6.0**	ns	ns	ns	ns	ns	ns
**−5.5**	ns	ns	ns	ns	ns	ns
**−5.0**	ns	ns	ns	ns	ns	ns
**log[AngII], M**	**ACC519T** **10^−50^ M vs. Control**		**BV6(K^+^)_2_** **10^−50^ M vs. Control**		**Candesartan** **10^−50^ M vs. Control**	
**−11.0**	ns		ns		ns	
**−10.5**	ns		ns		ns	
**−10.0**	ns		ns		*p* < 0.05	
**−9.5**	ns		*p* < 0.05		*p* < 0.001	
**−9.0**	ns		*p* < 0.0001		*p* < 0.0001	
**−8.5**	ns		*p* < 0.0001		*p* < 0.0001	
**−8.0**	ns		*p* < 0.0001		*p* < 0.0001	
**−7.5**	ns		*p* < 0.0001		*p* < 0.0001	
**−7.0**	ns		*p* < 0.0001		*p* < 0.0001	
**−6.5**	ns		*p* < 0.001		*p* < 0.0001	
**−6.0**	ns		ns		*p* < 0.05	
**−5.5**	ns		ns		ns	
**−5.0**	ns		ns		ns	
**log[AngII], M**	**ACC519T** **10^−60^ M vs. Control**		**BV6(K^+^)_2_** **10^−60^ M vs. Control**		**Candesartan** **10^−60^ M vs. Control**	
**−11.0**	ns		ns		ns	
**−10.5**	ns		ns		ns	
**−10.0**	ns		*p* < 0.05		ns	
**−9.5**	*p* < 0.01		*p* < 0.001		ns	
**−9.0**	*p* < 0.05		*p* < 0.0001		ns	
**−8.5**	*p* < 0.05		*p* < 0.0001		*p* < 0.05	
**−8.0**	*p* < 0.0001		*p* < 0.0001		*p* < 0.0001	
**−7.5**	*p* < 0.0001		*p* < 0.0001		*p* < 0.0001	
**−7.0**	*p* < 0.0001		*p* < 0.0001		*p* < 0.0001	
**−6.5**	*p* < 0.01		*p* < 0.0001		*p* < 0.0001	
**−6.0**	ns		ns		ns	
**−5.5**	ns		ns		ns	
**−5.0**	ns		ns		ns	

**Table 2 ijms-24-17559-t002:** Significant differences in contraction responses to PE dose–response effect after pre-treatment with various doses of drugs. Abbreviations: Abbreviations: ACC519T, benzilimidazole-*N*-biphenyl tetrazole; ACC519T(2), benzilimidazole *bis*-*N*,*N*′-biphenyl tetrazole; BV6(K^+^)_2_, 4-butyl-N,N0-bis[[20-2Htetrazol-5-yl)biphenyl-4-yl]methyl)imidazolium bromide; ns, no significance.

**log[PE], M**	**ACC519T** **10^−6.0^ M vs. Control**	**ACC519T(2)** **10^−6^ M vs. Control**	**BV6(K^+^)_2_** **10^−6^ M vs. Control**	**Candesartan** **10^−6^ M vs. Control**	**Nirmatrelvir** **10^−6^ M vs. Control**	**Telmisartan** **10^−6^ M vs. Control**
**−9.0**	ns	ns	ns	ns	ns	ns
**−8.5**	ns	ns	ns	ns	ns	ns
**−8.0**	ns	ns	ns	ns	ns	ns
**−7.5**	*p* < 0.01	ns	ns	ns	ns	ns
**−7.0**	*p* < 0.0001	ns	ns	ns	ns	ns
**−6.5**	*p* < 0.0001	ns	ns	ns	ns	ns
**−6.0**	*p* < 0.0001	ns	ns	ns	ns	ns
**−5.5**	*p* < 0.0001	ns	ns	ns	ns	ns
**−5.0**	*p* < 0.0001	ns	ns	ns	ns	ns
**log[PE], M**	**ACC519T** **10^−12^ M vs. Control**	**ACC519T(2)** **10^−12^ M vs. Control**	**BV6(K^+^)_2_** **10^−12^ M vs. Control**	**Candesartan** **10^−12^ M vs. Control**	**Nirmatrelvir** **10^−12^ M vs. Control**	**Telmisartan** **10^−12^ M vs. Control**
**−9.0**	ns	ns	ns	ns	ns	ns
**−8.5**	ns	ns	ns	ns	ns	ns
**−8.0**	ns	ns	ns	ns	ns	ns
**−7.5**	ns	ns	ns	ns	ns	ns
**−7.0**	ns	ns	ns	*p* < 0.01	ns	ns
**−6.5**	ns	*p* < 0.05	ns	*p* < 0.01	ns	ns
**−6.0**	ns	*p* < 0.05	ns	*p* < 0.01	ns	ns
**−5.5**	ns	ns	ns	ns	ns	ns
**−5.0**	ns	ns	ns	ns	ns	ns
**log[PE], M**	**ACC519T** **10^−18^ M vs. Control**	**ACC519T(2)** **10^−18^ M vs. Control**	**BV6(K^+^)_2_** **10^−18^ M vs. Control**	**Candesartan** **10^−18^ M vs. Control**	**Nirmatrelvir** **10^−18^ M vs. Control**	**Telmisartan** **10^−18^ M vs. Control**
**−9.0**	ns	ns	ns	ns	ns	ns
**−8.5**	ns	ns	ns	ns	ns	ns
**−8.0**	ns	ns	ns	ns	ns	ns
**−7.5**	ns	ns	ns	ns	ns	ns
**−7.0**	ns	*p* < 0.05	ns	ns	ns	ns
**−6.5**	ns	*p* < 0.01	ns	ns	ns	ns
**−6.0**	ns	*p* < 0.05	ns	ns	ns	ns
**−5.5**	ns	ns	ns	ns	ns	ns
**−5.0**	ns	ns	ns	ns	ns	ns
**log[PE], M**	**ACC519T** **10^−24^ M vs. Control**	**ACC519T(2)** **10^−24^ M vs. Control**	**BV6(K^+^)_2_** **10^−24^ M vs. Control**	**Candesartan** **10^−24^ M vs. Control**	**Nirmatrelvir** **10^−24^ M vs. Control**	**Telmisartan** **10^−24^ M vs. Control**
**−9.0**	ns	ns	ns	ns	ns	ns
**−8.5**	ns	ns	ns	ns	ns	ns
**−8.0**	ns	ns	ns	ns	ns	ns
**−7.5**	ns	ns	ns	ns	ns	ns
**−7.0**	ns	*p* < 0.05	ns	ns	ns	ns
**−6.5**	ns	*p* < 0.01	ns	ns	ns	ns
**−6.0**	ns	*p* < 0.0001	ns	ns	ns	ns
**−5.5**	ns	*p* < 0.01	ns	ns	ns	ns
**−5.0**	ns	*p* < 0.05	ns	ns	ns	ns
**log[PE], M**	**ACC519T** **10^−30^ M vs. Control**	**ACC519T(2)** **10^−30^ M vs. Control**	**BV6(K^+^)_2_** **10^−30^ M vs. Control**	**Candesartan** **10^−30^ M vs. Control**	**Nirmatrelvir** **10^−30^ M vs. Control**	**Telmisartan** **10^−30^ M vs. Control**
**−9.0**	ns	ns	ns	ns	ns	ns
**−8.5**	ns	ns	ns	ns	ns	ns
**−8.0**	ns	ns	ns	ns	ns	ns
**−7.5**	ns	ns	ns	ns	ns	ns
**−7.0**	ns	*p* < 0.01	ns	ns	ns	ns
**−6.5**	ns	*p* < 0.01	ns	ns	ns	ns
**−6.0**	ns	*p* < 0.05	ns	*p* < 0.01	ns	ns
**−5.5**	ns	ns	ns	*p* < 0.01	ns	ns
**−5.0**	ns	ns	ns	*p* < 0.01	ns	ns
**log[PE], M**	**ACC519T** **10^−40^ M vs. Control**	**ACC519T(2)** **10^−40^ M vs. Control**	**BV6(K^+^)_2_** **10^−40^ M vs. Control**	**Candesartan** **10^−40^ M vs. Control**	**Nirmatrelvir** **10^−40^ M vs. Control**	**Telmisartan** **10^−40^ M vs. Control**
**−9.0**	ns	ns	ns	ns	ns	ns
**−8.5**	ns	ns	ns	ns	ns	ns
**−8.0**	ns	ns	ns	ns	ns	ns
**−7.5**	ns	ns	ns	ns	ns	ns
**−7.0**	ns	ns	ns	ns	ns	ns
**−6.5**	*p* < 0.05	ns	ns	*p* < 0.05	ns	*p* < 0.05
**−6.0**	*p* < 0.0001	ns	ns	*p* < 0.01	ns	*p* < 0.01
**−5.5**	*p* < 0.0001	ns	ns	*p* < 0.01	ns	*p* < 0.01
**−5.0**	*p* < 0.0001	ns	ns	*p* < 0.01	ns	*p* < 0.01
**log[PE], M**	**ACC519T** **10^−50^ M vs. Control**		**BV6(K^+^)_2_** **10^−50^ M vs. Control**		**Candesartan** **10^−50^ M vs. Control**	
**−9.0**	ns		ns		ns	
**−8.5**	ns		ns		ns	
**−8.0**	ns		ns		*p* < 0.0001	
**−7.5**	ns		ns		*p* < 0.0001	
**−7.0**	ns		ns		*p* < 0.001	
**−6.5**	ns		ns		*p* < 0.001	
**−6.0**	ns		*p* < 0.05		*p* < 0.05	
**−5.5**	ns		*p* < 0.05		ns	
**−5.0**	ns		ns		ns	
**log[PE], M**	**ACC519T** **10^−60^ M vs. Control**		**BV6(K^+^)_2_** **10^−60^ M vs. Control**		**Candesartan** **10^−60^ M vs. Control**	
**−9.0**	ns		ns		ns	
**−8.5**	ns		ns		ns	
**−8.0**	ns		ns		ns	
**−7.5**	ns		ns		ns	
**−7.0**	ns		ns		ns	
**−6.5**	ns		ns		*p* < 0.05	
**−6.0**	ns		ns		*p* < 0.05	
**−5.5**	ns		ns		ns	
**−5.0**	ns		ns		ns	

**Table 3 ijms-24-17559-t003:** Significant differences in contraction responses to AngA dose–response effect after pre-treatment with various doses of ARBs. Abbreviations: Abbreviations: ACC519T, benzilimidazole-*N*-biphenyl tetrazole; ACC519T(2), benzilimidazole *bis*-*N*,*N*′-biphenyl tetrazole; BV6(K^+^)_2_, 4-butyl-N,N0-bis[[20-2Htetrazol-5-yl)biphenyl-4-yl]methyl)imidazolium bromide; ns, no significance.

**log[AngA], M**	**ACC519T 10^−6^ M vs. Control**	**ACC519T(2) 10^−6^ M vs. Control**	**BV6(K^+^)_2_ 10^−6^ M vs. Control**
**−11.0**	ns	ns	ns
**−10.5**	ns	ns	ns
**−10.0**	ns	ns	ns
**−9.5**	ns	*p* < 0.05	*p* < 0.05
**−9.0**	ns	*p* < 0.0001	*p* < 0.0001
**−8.5**	ns	*p* < 0.0001	*p* < 0.0001
**−8.0**	*p* < 0.01	*p* < 0.0001	*p* < 0.0001
**−7.5**	*p* < 0.0001	*p* < 0.0001	*p* < 0.0001
**−7.0**	*p* < 0.0001	*p* < 0.0001	*p* < 0.0001
**−6.5**	*p* < 0.0001	*p* < 0.0001	*p* < 0.0001
**−6.0**	*p* < 0.0001	*p* < 0.0001	*p* < 0.0001
**−5.5**	*p* < 0.0001	*p* < 0.0001	*p* < 0.0001
**−5.0**	ns	*p* < 0.001	ns
**log[AngA], M**	**ACC519T 10^−12^ M vs. Control**	**ACC519T(2) 10^−12^ M vs. Control**	**BV6(K^+^)_2_ 10^−12^ M vs. Control**
**−11.0**	ns	ns	ns
**−10.5**	ns	ns	ns
**−10.0**	ns	ns	ns
**−9.5**	ns	*p* < 0.01	*p* < 0.05
**−9.0**	*p* < 0.001	*p* < 0.0001	*p* < 0.0001
**−8.5**	*p* < 0.0001	*p* < 0.0001	*p* < 0.0001
**−8.0**	*p* < 0.0001	*p* < 0.0001	*p* < 0.0001
**−7.5**	*p* < 0.0001	*p* < 0.0001	*p* < 0.0001
**−7.0**	*p* < 0.0001	*p* < 0.0001	*p* < 0.0001
**−6.5**	*p* < 0.0001	*p* < 0.0001	*p* < 0.0001
**−6.0**	*p* < 0.0001	*p* < 0.0001	*p* < 0.0001
**−5.5**	*p* < 0.0001	*p* < 0.0001	*p* < 0.0001
**−5.0**	ns	*p* < 0.01	*p* < 0.05
**log[AngA], M**	**ACC519T 10^−18^ M vs. Control**	**ACC519T(2) 10^−18^ M vs. Control**	**BV6(K^+^)_2_ 10^−18^ M vs. Control**
**−11.0**	ns	ns	ns
**−10.5**	ns	ns	ns
**−10.0**	ns	ns	ns
**−9.5**	ns	*p* < 0.05	ns
**−9.0**	*p* < 0.001	*p* < 0.0001	*p* < 0.0001
**−8.5**	*p* < 0.0001	*p* < 0.0001	*p* < 0.0001
**−8.0**	*p* < 0.0001	*p* < 0.0001	*p* < 0.0001
**−7.5**	*p* < 0.0001	*p* < 0.0001	*p* < 0.0001
**−7.0**	*p* < 0.0001	*p* < 0.0001	*p* < 0.0001
**−6.5**	*p* < 0.0001	*p* < 0.0001	*p* < 0.0001
**−6.0**	*p* < 0.0001	*p* < 0.0001	*p* < 0.0001
**−5.5**	*p* < 0.0001	*p* < 0.0001	*p* < 0.0001
**−5.0**	ns	*p* < 0.01	*p* < 0.05
**log[AngA], M**	**ACC519T 10^−24^ M vs. Control**	**ACC519T(2) 10^−24^ M vs. Control**	**BV6(K^+^)_2_ 10^−24^ M vs. Control**
**−11.0**	ns	ns	ns
**−10.5**	ns	ns	ns
**−10.0**	ns	ns	ns
**−9.5**	ns	*p* < 0.01	ns
**−9.0**	*p* < 0.0001	*p* < 0.0001	*p* < 0.0001
**−8.5**	*p* < 0.0001	*p* < 0.0001	*p* < 0.0001
**−8.0**	*p* < 0.0001	*p* < 0.0001	*p* < 0.0001
**−7.5**	*p* < 0.0001	*p* < 0.0001	*p* < 0.0001
**−7.0**	*p* < 0.0001	*p* < 0.0001	*p* < 0.0001
**−6.5**	*p* < 0.0001	*p* < 0.0001	*p* < 0.0001
**−6.0**	*p* < 0.0001	*p* < 0.0001	*p* < 0.0001
**−5.5**	*p* < 0.0001	*p* < 0.0001	*p* < 0.0001
**−5.0**	ns	*p* < 0.001	*p* < 0.05
**log[AngA], M**	**ACC519T 10^−30^ M vs. Control**	**ACC519T(2) 10^−30^ M vs. Control**	**BV6(K^+^)_2_ 10^−30^ M vs. Control**
**−11.0**	ns	ns	ns
**−10.5**	ns	ns	ns
**−10.0**	ns	ns	ns
**−9.5**	ns	*p* < 0.01	*p* < 0.05
**−9.0**	*p* < 0.0001	*p* < 0.0001	*p* < 0.0001
**−8.5**	*p* < 0.0001	*p* < 0.0001	*p* < 0.0001
**−8.0**	*p* < 0.0001	*p* < 0.0001	*p* < 0.0001
**−7.5**	*p* < 0.0001	*p* < 0.0001	*p* < 0.0001
**−7.0**	*p* < 0.0001	*p* < 0.0001	*p* < 0.0001
**−6.5**	*p* < 0.0001	*p* < 0.0001	*p* < 0.0001
**−6.0**	*p* < 0.0001	*p* < 0.0001	*p* < 0.0001
**−5.5**	*p* < 0.0001	*p* < 0.0001	*p* < 0.0001
**−5.0**	ns	*p* < 0.001	ns
**log[AngA], M**	**ACC519T 10^−40^ M vs. Control**	**ACC519T(2) 10^−40^ M vs. Control**	**BV6(K^+^)_2_ 10^−40^ M vs. Control**
**−11.0**	ns	ns	ns
**−10.5**	ns	ns	ns
**−10.0**	ns	ns	ns
**−9.5**	ns	*p* < 0.05	ns
**−9.0**	*p* < 0.001	*p* < 0.0001	*p* < 0.0001
**−8.5**	*p* < 0.0001	*p* < 0.0001	*p* < 0.0001
**−8.0**	*p* < 0.0001	*p* < 0.0001	*p* < 0.0001
**−7.5**	*p* < 0.0001	*p* < 0.0001	*p* < 0.0001
**−7.0**	*p* < 0.0001	*p* < 0.0001	*p* < 0.0001
**−6.5**	*p* < 0.0001	*p* < 0.0001	*p* < 0.0001
**−6.0**	*p* < 0.0001	*p* < 0.0001	*p* < 0.0001
**−5.5**	*p* < 0.0001	*p* < 0.0001	*p* < 0.0001
**−5.0**	ns	*p* < 0.001	*p* < 0.05

## Data Availability

The data are not publicly available due to commercial value. Interested parties can email corresponding authors.

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
