# Peer review of "Existence of Quantum Pharmacology in Sartans: Evidence in Isolated Rabbit Iliac Arteries"

_ijms, 2023, doi:10.3390/ijms242417559_

Round 1

Reviewer 1 Report

Comments and Suggestions for Authors

1. The abstract and introduction lack logical coherence in addressing the identified issues. It is essential to clearly demonstrate how this study contributes to addressing these existing problems.

2. When first introduced, please spell out "MAP" (Mean Arterial Pressure) and "ARB" (Angiotensin II Receptor Blockers) for better clarity.

3. Ensure that a control group is included in Figure 1 for comprehensive comparison.

4. In the paragraph spanning from line 139 to 155, consider reducing the number of p-values presented to enhance the paragraph's conciseness and efficiency. Apply a similar approach throughout the manuscript.

5. Provide an explanation for the varying "n" numbers in Figure 3 across different groups and clarify the control group.

6. Streamline the discussion section to improve efficiency and conciseness in delivering the key points.

7. In the conclusion, clearly state the problems addressed by the study and how the findings could be applied to guide medication.

Author Response

Reviewer #1

  1. The abstract and introduction lack logical coherence in addressing the identified issues. It is essential to clearly demonstrate how this study contributes to addressing these existing problems.

Thank you for providing this comment. We also agree that the introduction and abstract needs to clearly demonstrate how this study contributes to addressing these existing problems. The abstract now reads as:

Line 21-42: Quantum pharmacology introduces theoretical models to describe the possibility of ultra-high dilutions to produce biological effects, which may help to explain the placebo effect observed in hypertensive clinical trials. To determine this within physiology and to evaluate novel ARBs, we tested the ability of known angiotensin II receptor blockers (ARBs) (candesartan and telmisartan) used to treat hypertension and other cardiovascular diseases, as well as novel ARBs (benzimidazole-N-biphenyl tetrazole (ACC519T), benzimidazole-bis-N,N’-biphenyl tetrazole (ACC519T(2)) and 4-butyl-N,N0-bis[20-2Htetrazol-5-yl)bipheyl-4-yl]methyl)imidazolium bromide (BV6(K+)2), and nirmatrelvir (the active ingredient in Paxlovid) to modulate vascular contraction in iliac rings from healthy male New Zealand White rabbits in responses to various vasopressors (angiotensin A, angiotensin II and phenylephrine). Additionally, the hemodynamic effect of ACC519T and telmisartan on mean arterial pressure in conscious rabbits was determined, while the ex vivo ability of BV6(K+)2 to activate angiotensin converting enzyme-2 (ACE2) was also shown. We show that commercially available and novel ARBs can modulate contraction responses at ultra-high dilutions to different vasopressors. ACC519T produced dose-dependent reduction in rabbit mean arterial pressure while BV6(K+)2 significantly increased ACE2 metabolism. The ability of ARBs to inhibit contraction responses even at ultra-low concentrations provides evidence of the existence of quantum pharmacology. Furthermore, the ability of ACC519T and BV6(K+)2 to modulate blood pressure and ACE2 activity, respectively, indicate their therapeutic potential against hypertension.

 We have now added the following paragraphs and sentences:

Line 46-66: Placebo groups are an integral component in randomized controlled drug trials to allow differentiation between the specific effects of a pharmaceutical from nonspecific changes in symptomology caused by a pharmacologically inactive substance [1]. Therefore, the effects observed in the experimental drug group are required to be substantial in comparison to those in the placebo group to establish drug efficacy and justify its use [1]. However, reports from drug trials demonstrate that both statistically and clinically significant improvements in symptoms have been observed in placebo groups, a concept defined as the “placebo effect” [1]. Incredibly, in contrast to the placebo effect, adverse or even toxic consequences have also been reported in placebo groups, a phenomenon known as the “nocebo effect” [2]. The paradox of these phenomena lies within the widely accepted dogma that an inert substance by definition is unable to produce an effect; neither positive or negative [2]. However, there is compelling evidence from pharmacological studies indicating that placebos mimic the action of drugs through shared biochemical pathways by associating and activating the same receptor [3]. In attempt to explain the complex underlying mechanisms of the placebo effect multiple theories, such as natural course of disease, fluctuations in severity of symptoms, response bias involving patient reporting, other concurrent treatments, psychosocial expectation responses, and quantum physics and mechanics, have been hypothesized [2, 4]. The latter of which suggests that information entanglement driven by human intention can alter the physical and biochemical properties and subsequently the physiological response of a placebo on a molecular and atomic level [4].

Line 110-129: Hypertension is a major health concern, contributing significantly to cardiovascular morbidity and mortality [29, 30]. Hypertension is often treated pharmacologically and has contributed to a major global reduction in the incidence of cardiovascular disease [31]; however, despite this, inadequate blood pressure control persists. Due to reduced patient sensitivity to antihypertensive therapy, overmedication is relevant [32]. Patients often require at least two or more drugs for adequate blood pressure control, which inevitably increase the risk of adverse effects such as hypotension [33, 34]. Therefore, the need for novel, more effective pharmaceuticals to treat hypertension is desperately needed. Sartans are a family of ARBs used to treat hypertension and other cardiovascular related diseases by targeting the renin angiotensin system (RAS) from vasoconstrictive hormones, AngII and angiotensin A (AngA). Clinical trials evaluating the efficacy of novel anti-hypertensives have observed reduction in systolic and diastolic blood pressure and overall lowering of blood pressure in participants in the placebo group, suggesting that the cardiovascular system may be sensitive to placebo mechanisms induced by quantum entanglement, physics and pharmacology [1, 35]. We have previously designed a series of novel sartans, based on the structure of losartan [36, 37], called bisartans, which appear to be superior ARBs when compared to commercially available sartans [38]. Bisartans exhibit dual antihypertensive and antiviral abilities through angiotensin type 1 receptor (AT1R) blockage and inhibiting entry of severe-acute respiratory syndrome coronavirus 2 (SARS-CoV-2) spike protein through destabilization of the angiotensin converting enzyme-2 (ACE2) receptor binding domain complex, and thus preventing coronavirus 2019 (COVID-19) [38-41].

Line 133-146: The present study aimed to aid in the understanding of the placebo effect by investigating the possible existence of quantum entanglement in isolated rabbit iliac arteries and evaluating the ability of our newly synthesized bisartans to behave as novel ARBs for the potential treatment of hypertension. Arteries were incubated with known (candesartan and telmisartan) and novel (benzimidazole-N-biphenyl tetrazole (ACC519T), benzimidazole-bis-N-N’-biphenyl tetrazole (ACC519T(2)) and 4-butyl-N,N0-bis[20-2Htetrazol-5-yl)bipheyl-4-yl]methyl)imidazolium bromide (BV6(K+)2) (Figure 1) ARBs synthesized by our group and nirmatrelvir (COVID-19 medication) at extremely low concentrations prepared by serial dilution in pure water in order to investigate the validity of ultra-high dilutions to provide biological effects through their ability to antagonize the AT1R in response to various vasopressive compounds (i.e., AngII, phenylephrine (PE) and AngA). Additionally, the ability of BV6(K+)2 to interact with ACE2 at ultra-high dilutions was also investigated, while the novel drug, ACC519T, and telmisartan was also injected into conscious rabbits to determine their haemodynamic effect on mean arterial pressure (MAP).

  1. When first introduced, please spell out "MAP" (Mean Arterial Pressure) and "ARB" (Angiotensin II Receptor Blockers) for better clarity.

Thank you for bringing this to our attention; the manuscript has been read in its entirety to make sure all abbreviations have first been introduced.

  1. Ensure that a control group is included in Figure 1 for comprehensive comparison.

            Thank you for this comment, we agree that a control group is important. Unfortunately, due to the COVID-19 pandemic our rabbit suppliers (Flinders University, South Australia) closed their breeding colonies over a year ago before the end of our study and are no longer breeding rabbits. We have been unable to find other supplier and thus we are not able to provide MAP recordings in control (untreated) rabbits.

  1. In the paragraph spanning from line 139 to 155, consider reducing the number of p-values presented to enhance the paragraph's conciseness and efficiency. Apply a similar approach throughout the manuscript.

We greatly appreciate this comment, as we there is a lot of data, we want to ensure that it is concise and efficiently presented. We have now only included the most significant p value of each drug in the written results sections.

  1. Provide an explanation for the varying "n" numbers in Figure 3 across different groups and clarify the control group.

Thank you for this comment. Varying “n” numbers in contraction responses to AngII was a result of inconsistencies in the number of iliac rings retrievable from each rabbit while rings that did not response to KPSS were omitted from the study. Additionally, as the experiment was conducted between two batches of rabbits, the first batch (n=6) was used to determine traditional response effects of ARBs while the second batch (n=4) was used to confirm effects for low dose ARB effects, thus contributing to the varying “n” numbers between higher and lower ARB incubation doses. To make this clearer to reader we have now added the following Lines:

Line 501-504: The first batch of rabbits (n = 6) were used for a preliminary pilot to determine traditional Emax and evaluate the ability of experimental ARBs to alter contraction responses in iliac artery rings. The second batch of rabbits (n = 4) were used to confirm the ability of ARBs to reduce contraction at low doses.

Line 584-587: Due to some doses being performed in both batches of rabbits and others only in the first or second batch, differences in the number of iliac rings retriable from each rabbit and rings that did not response to KPSS being omitted from the study an n = 10-3 was observed across both rabbit batches.

To add more clarity on control groups we have added to following lines to the methodology and results sections, respectively:

Figure 3, 4 and 5: Contraction responses to AngII dose-response in rabbit iliac arteries left to rest (control) or pre-treated with various doses of commercial and experimental ARBs.

Line 570-572: Rings serving as control groups were not incubated with any drug and left to rest for 10 min, while in ARB experimental groups rings were incubated with…”

  1. Streamline the discussion section to improve efficiency and conciseness in delivering the key points.

Thank-you for this comment. We also agree that the discussion needs to be improved and have condensed this section to focus on key points.

  1. In the conclusion, clearly state the problems addressed by the study and how the findings could be applied to guide medication.

Thank you for this comment and we have now condensed and changed the conclusions:

Line 617-640: Herein, we demonstrate the ability of extraordinarily low doses of both our novel and commonly prescribed ARBs to inhibit AngII and AngA contraction responses and augment PE contraction responses in isolated rabbit iliac arteries. Incredibly, we also show that a low dose of the ARB ACC519T is able to lower MAP in conscious rabbits. Receptor responses can be extremely complex due to electromagnetic activity, "cross interactions" among ligands, receptors, and intracellular signalling mechanisms. We speculate the mechanism of ultra-low dilutions to be aligned with quantum effects of ARBs diluted in pure water where their antagonistic effects may be retained through quantum electromagnetic effects to produce receptor interaction, which may aid in explaining the placebo effect that is seen in randomized controlled trials of different drugs. Moreover, the ability of these our newly synthesized bisartans and commercially available ARBs to have an effect at extremely low concentrations challenges the current dosage given to hypertensive patients. Additionally, augmentation of PE responses in the presence of ACC519T may be due to receptor "cross talk" wherein deactivation of the AngII receptor is offset by up regulation of the α adrenergic receptor response mechanism, possibly via interlinked second messenger systems. Likewise, the reversal of the dose-response effect of ACC519T(2), in which low doses (10-40 M) block AngII but high doses (l0-6 M) do not, contrasts with the ACC519T(2) blockade of AngA at all doses (which is similar to ACC519T and BV6(K+)2) and suggests a different receptor mechanism for AngA. Future in silico and molecular dynamic docking studies may be useful in investigating the different mechanism of ARB receptor interaction. It is essential that further investigation on the underlying mechanism of our results is performed and that other research groups can replicate our finding in order to validate the effects of ultra-high dilutions on various biological systems.

Reviewer 2 Report

Comments and Suggestions for Authors

Comments to the Author:

In the manuscript entitled Existence of quantum pharmacology in sartans: evidence in isolated rabbit iliac arteries authors have tested the ability of known angiotensin II receptor blockers (ARBs) (candesartan and telmisartan) used to treat  hypertension and other cardiovascular diseases, as well as novel ARBs (benzimidazole-N-biphenyl  tetrazole (ACC519T), benzimidazole-bis-N,N’-biphenyl tetrazole (ACC519T(2)) and 4-butyl-N,N0- bis[20-2H-tetrazol-5-yl)bipheyl-4-yl]methyl)imidazolium bromide (BV6(K+)2), and nirmatrelvir (the  active ingredient in Paxlovid) to modulate vascular contraction in iliac rings from healthy male New  Zealand White rabbits in responses to various vasopressors (angiotensin A, angiotensin II and phenylephrine). Also the ability of ACC519T and BV6(K+)2 to modulate blood pressure and ACE2 activity, respectively, indicate their therapeutic potential against hypertension.

I have read the manuscript carefully and found that the manuscript is well written and explains the results very nicely.

However, I have following queries.

1.      In figure 1, Structure D should be corrected.

2.      Line 316, spellings of confirmation should be corrected ad it should be conformation.

3.      Symbols for alpha and beta should be used.

4.      Line 608, M should be corrected to M.

With these changes, manuscript can be accepted for publication.

Comments on the Quality of English Language

Minor changes required

Author Response

Reviewer #2

  1. In figure 1, Structure D should be corrected.

            Thank you for pointing this out, we have now corrected the chemical structure of candesartan in Figure 1D.

  1. Line 316, spellings of confirmation should be corrected ad it should be conformation.

Thank you very much for pointing out this spelling mistake. We have now corrected this mistake and have carefully read the manuscript in its entirety to ensure any other spelling and grammatical errors have been corrected.

  1. Symbols for alpha and beta should be used.

            We appreciate this suggestion and we have replaced alpha and beta with the correct Greek symbol throughout the manuscript.

  1. Line 608, M should be corrected to M.

            Thank you for bringing this to our attention. M is no longer in superscript and the manuscript has been checked for any other sub- or superscript errors.

Reviewer 3 Report

Comments and Suggestions for Authors

The manuscript is well written and introduces the topic of new research but needs to improve in several areas; these issues must be addressed to make it attractive for the journal readers.

Point 1: The study's objectives and rationale should be clearly stated?

Point 2: The application and methodology should be reported sufficiently to allow for its replicability and/or reproducibility?

Point 3: Why did the authors determine the ability of angiotensin II receptor blockers? 

Point 4: Why did the authors choose white rabbits instead of mice or rats? Explain.

Point 5: Why did this study compare the ACC519T and telmisartan? Clarify.

Point 6: The introduction must be concise to grasp and attract the readers.

Point 7: Why did the authors choose telmisartan over other standard drugs?

Point 8: There is so much difference in BV6(K+ )2 (10-6, 10-24, and 10-60) figure-6. Explain.

Point 9: There is a significant difference between Table 3 and Figure 6. Explain.

Point 10: Why were the drug incubations and isometric tension myography studies conducted?

Author Response

Reviewer #3

  1. The study's objectives and rationale should be clearly stated?

            We greatly appreciate this and agree that our rationale was not clearly defined. We have now added the following:

Line 133-136: The present study aimed to aid in the understanding of the placebo effect by investigating the possible existence of quantum entanglement in isolated rabbit iliac arteries and evaluating the ability of our newly synthesized bisartans to behave as novel ARBs for the potential treatment for hypertension.

  1. The application and methodology should be reported sufficiently to allow for its replicability and/or reproducibility?

            Thank you for this comment, we have now gone through section 4 materials and methods to make sure that all materials and their corresponding catalogue number are correct and have elaborated on some of the methods section to sufficient report what was done allowing for replicability and reproducibility. To improve replicability and reproducibility of our methodology we have added:

Line 584-590: Due to some doses being performed in both batches of rabbits and others only in the first or second batch, differences in the number of iliac rings retriable from each rabbit and rings that did not response to KPSS being omitted from the study an n = 10-3 was observed across both rabbit batches. Preparation of ARB concentration occurred through serial dilution in which 1:1000 dilutions were performed in individual plastic Eppendorf tubes until the desired concentration was reached. Solutions were vortexed between each concurrent dilution.

Line 607-610: Identification of peptides and quantification was based upon the elution time of synthetic standards and known amounts of AngII and Ang (1-7). The mean absorbance of replicates between each samples was calculated, and the standard curve was used to interpolate values of both peptides.

Line 612-617: GraphPad prism (version 9.5.1) was utilized for statistical analysis of isometric tension, MAP data, and ACE2 activity. A one-way ANOVA followed by Dunnett’s (ACE2) or Sidak’s post hoc was utilized to determine significance in ACE2 activity and MAP studies, respectively. A two-way ANOVA followed by Sidak’s post hoc was used to determine significance in isometric tension myography results. The significant p-value was set at p < 0.05, and all data are represented as mean ± standard error of mean (SEM).

  1. Why did the authors determine the ability of angiotensin II receptor blockers? 

Thank you for this question. Our laboratory has immense knowledge in the field of cardiovascular diseases, including hypertension, and we have synthesized our own novel antihypertensives, called bisartans. Moreover, our bisartans have shown dual antiviral and antihypertensive abilities, and appear to be superior to commercially available sartans. Due to this we wanted to extend our research by evaluating the ability of these bisartans to not only reduce AngII-mediated contraction at extremely low doses and be a potential novel treatment for hypertension, but also to determine their ability to effect contraction to other vasopressors.

  1. Why did the authors choose white rabbits instead of mice or rats? Explain.

This is an excellent question, and we greatly appreciate the opportunity to explain the reasoning for using rabbits as our experimental animal in this study. The reasons for using rabbits over rodents include: (a) medications that target different components of the cardiovascular system, such as some statins and sartans, were first discovered and evaluated in rabbit models before proceeding to human clinical trials; (b) rodents lack responses to AngII in conduit arteries (e.g., carotid arteries and thoracic and abdominal aorta), which may be explained by rodents having AT1R subtype A and B, where humans and rabbits express AT1R with no subtypes; and (c) expression of AT1R subtypes A and B are poorly expressed in rodent conduit arteries but are abundantly expressed in conduit arteries in humans and rabbits. We have now added the following statement to explain to the reader why rabbits were used in this study:

Line 510-522: Rabbits were chosen for this study as they share closer phylogenetic resemblance to humans and pioneering studies derived from rabbit experiments support their use as a more reliable model for studying cardiovascular physiology, mechanisms, and pathology than laboratory rodents (e.g., mice, rats, and guinea pigs), which have limited translational impact [101]. Importantly, rabbits and humans elicit an almost identical vascular response to AngII in conduit arteries. An often-ignored problem inherent to the use of rodent models is the existence of two AT1R isoforms, namely AT1R subtype A and B [102], the latter of which is predominantly responsible for mediating AngII vasoconstriction responses [103]. However, we [104] and others [103, 105] have shown the inability or reduced capacity of AngII to stimulate contraction responses in rodent blood vessels, including the thoracic and abdominal aorta, and carotid and brachiocephalic arteries. This may be explained by reduced sensitivity to AngII, desensitization of AT1R subtypes subsequent to successive AngII exposure and poor expression profiles in blood vessels [106].

  1. Why did this study compare the ACC519T and telmisartan? Clarify.

            Thank for this question. Our newly designed and synthesized drug, ACC519T, closely resembles the chemical structure of the commercially available ARB telmisartan. Thus, due to the similarity in structure, these two drugs were chosen for comparison in live rabbit MAP studies. Please refer to comment 7 for the statement that has been added to section 4.2.

  1. The introduction must be concise to grasp and attract the readers.

            Thank you this, we agree that the introduction was lacking concepts that the readers required to understand our study. We have now added important information to bridge these gaps.

Line 46-66: Placebo groups are an integral component in randomized controlled drug trials to allow differentiation between the specific effects of a pharmaceutical from nonspecific changes in symptomology caused by a pharmacologically inactive substance [1]. Therefore, the effects observed in the experimental drug group are required to be substantial in comparison to those in the placebo group to establish drug efficacy and justify its use [1]. However, reports from drug trials demonstrate that both statistically and clinically significant improvements in symptoms have been observed in placebo groups, a concept defined as the “placebo effect” [1]. Incredibly, in contrast to the placebo effect, adverse or even toxic consequences have also been reported in placebo groups, a phenomenon known as the “nocebo effect” [2]. The paradox of these phenomena lies within the widely accepted dogma that an inert substance by definition is unable to produce an effect; neither positive or negative [2]. However, there is compelling evidence from pharmacological studies indicating that placebos mimic the action of drugs through shared biochemical pathways by associating and activating the same receptor [3]. In attempt to explain the complex underlying mechanisms of the placebo effect multiple theories, such as natural course of disease, fluctuations in severity of symptoms, response bias involving patient reporting, other concurrent treatments, psychosocial expectation responses, and quantum physics and mechanics, have been hypothesized [2, 4]. The latter of which suggests that information entanglement driven by human intention can alter the physical and biochemical properties and subsequently the physiological response of a placebo on a molecular and atomic level [4].

Line 110-129: Hypertension is a major health concern, contributing significantly to cardiovascular morbidity and mortality [29, 30]. Hypertension is often treated pharmacologically and has contributed to a major global reduction in the incidence of cardiovascular disease [31]; however, despite this, inadequate blood pressure control persists. Due to reduced patient sensitivity to antihypertensive therapy, overmedication is relevant [32]. Patients often require at least two or more drugs for adequate blood pressure control, which inevitably increase the risk of adverse effects such as hypotension [33, 34]. Therefore, the need for novel, more effective pharmaceuticals to treat hypertension is desperately needed. Sartans are a family of ARBs used to treat hypertension and other cardiovascular related diseases by targeting the renin angiotensin system (RAS) from vasoconstrictive hormones, AngII and angiotensin A (AngA). Clinical trials evaluating the efficacy of novel anti-hypertensives have observed reduction in systolic and diastolic blood pressure and overall lowering of blood pressure in participants in the placebo group, suggesting that the cardiovascular system may be sensitive to placebo mechanisms induced by quantum entanglement, physics and pharmacology [1, 35]. We have previously designed a series of novel sartans, based on the structure of losartan [36, 37], called bisartans, which appear to be superior ARBs when compared to commercially available sartans [38]. Bisartans exhibit dual antihypertensive and antiviral abilities through angiotensin type 1 receptor (AT1R) blockage and inhibiting entry of severe-acute respiratory syndrome coronavirus 2 (SARS-CoV-2) spike protein through destabilization of the angiotensin converting enzyme-2 (ACE2) receptor binding domain complex, and thus preventing coronavirus 2019 (COVID-19) [38-41].

  1. Why did the authors choose telmisartan over other standard drugs?

            Thank you for this question, we agree that we need to explain why we have chosen both candesartan and telmisartan. The reason as to why telmisartan and candesartan were chosen over other commonly prescribed sartans/antihypertensives is due to them sharing similar chemical structures with our newly synthesized bisartans – candesartan is chemically similar to BV6(K+)2 and telmisartan shares similar chemical structure to ACC519T and ACC519T(2). Thus, they provide positive controls for our drugs that are commercially available, commonly prescribed and have a well-established/known effect. To explain this to the reader we have now added the following statement to section 4.2:

Line 488-492: Due to the similar chemical structure shared between ACC519T, ACC519T(2) and telmisartan, and BV6(K+)2 and candesartan, these commercially available and commonly prescribed pharmaceuticals were used in this experiment as positive controls with known effects to validate the effectiveness of our newly synthesized ARBs.

  1. There is so much difference in BV6(K+ )2 (10-6, 10-24, and 10-60) figure-6. Explain.

We appreciate this question. At 10-6 M BV6(K+)2 was able to significantly increases ACE2 activity by reducing AngII and subsequently increasing Ang(1-7) concentration while at ultra-diluted concentrations of 10-24 M and 10-60 M BV6(K+)2 had no effect on ACE2 activity. This absence of effects in ultra-diluted doses of BV6(K+)2 is inconsistent with AngII dose-response effects. We hypothesize that ultra-high dilution-receptor interactions may be related to enzyme/receptor structure rather than synthetic ARB drugs as an explanation for the absence in effect of low-dose BV6(K+)2 on ACE2 activity. To make this more apparent we have added significance to Figure 6 between BV6(K+)2 doses and added the following segment:

Line 286-292: We report the ability of bisartan compound BV6(K+)2 to significantly increase ACE2 activity at 10-6 M, as seen by the metabolism of AngII into Ang(1-7). However, in contrast to AngII-dose-response results, at ultra-diluted concentrations, BV6(K+)2 incubation with ACE2 failed to increase enzymatic activity. Therefore, we hypothesize that the metalloenzyme ACE2 and G protein-coupled receptor AT1R may rely on independent ultra-high dilution-receptor interactions that are related to enzyme/receptor structure.

  1. There is a significant difference between Table 3 and Figure 6. Explain.

            Thank you for this comment. We were wondering if we could please have some clarity. Table 3 refers to the significant differences of Phen Dose-responses of the experimental drugs and Figure 6 refers to ACE2 activity, as the two aren't related we were wondering if we could please have some clarification as to what is meant by the comment.

  1. Why were the drug incubations and isometric tension myography studies conducted?

              We greatly appreciate this question and the opportunity to clarify to the reader as to why isometric tension myography studies are imperative to our study methodology. Isometric tension myography is a gold standard functional test used to determine the pharmacodynamic effects that a pharmaceutical has on blood vessel function. Importantly, these studies allow a real-time trace to determine the effect that a drug has on the ability of blood vessels to relax or contract to different vasoactive peptides. Our laboratory has extensive expertise in the use of isometric tension myography studies, and we have 120 independent organ bath chambers, allowing us to test the vasoactive responses of multiple drugs in response to a multitude of different vasodilators and vasoconstrictors. The following statement has been added to situate the reader as to why isometric tension studies were used:

Line 558-563: Isometric tension myography is a gold standard functional test used to determine the pharmacodynamic effects that a pharmaceutical has on blood vessel function. Importantly, these studies allow a real-time trace to determine the effect that a drug has on the ability of blood vessels to relax or contract to different vasoactive peptides, providing an ex vivo understanding of isolated and potentially systemic effects that a drug may have on blood vessel function.

Round 2

Reviewer 1 Report

Comments and Suggestions for Authors

The author has undoubtedly put forth significant effort to improve the quality of the manuscript and address my concerns. It's unfortunate to observe that a lack of control stems from the supplier's discontinuation, which was triggered by the impact of COVID.

Comments on the Quality of English Language

The manuscript's English proficiency is adequate for communication.

Reviewer 3 Report

Comments and Suggestions for Authors

Most of the suggestions are incorporated in manuscript. It could be published.